# Distributionally Robust Reinforcement Learning from Human Feedback

**Debmalya Mandal** [1]   **Paulius Sasnauskas** [2][3][†]   **Goran Radanović** [4]

## Abstract

Reinforcement learning from human feedback (RLHF) has evolved to be one of the main methods for fine-tuning large language models (LLMs). However, existing RLHF methods are non-robust, and their performance deteriorates if the downstream task differs significantly from the preference dataset used in fine-tuning. In order to mitigate this problem, we introduce a distributionally robust RLHF for fine-tuning LLMs. In particular, our goal is to ensure that a fine-tuned model retains its performance even when the distribution of prompts significantly differs from the distribution encountered during fine-tuning. We formulate distributionally robust optimization (DRO) version of two popular fine-tuning methods – (1) reward-based RLHF and (2) reward-free DPO (direct preference optimization). We propose a minibatch gradient descent based algorithms for both of them, and theoretically prove convergence guarantees for the algorithms. Subsequently, we evaluate our algorithms on an out-of-distribution (OOD) task by first training the model on the Unified-Feedback dataset and evaluating its performance on two different datasets. The experimental results show that our robust training improves the accuracy of the learned reward models on average, and markedly on some tasks, such as reasoning. Furthermore, we show that the robust versions of policy optimization methods, similarly improve performance on OOD tasks.

---

[†]This work was done as a part of an internship project at MPI-SWS. [1]Department of Computer Science, University of Warwick, Coventry, United Kingdom [2]Department of Computing Science, University of Alberta, Edmonton, Canada [3]Alberta Machine Intelligence Institute (Amii), Edmonton, Canada [4]Max Planck Institute for Software Systems, Saarbrücken, Germany. Correspondence to: Debmalya Mandal <Debmalya.Mandal@warwick.ac.uk>.

*Proceedings of the 43rd International Conference on Machine Learning*, Seoul, South Korea. PMLR 306, 2026. Copyright 2026 by the author(s).

## 1. Introduction

Reinforcement learning with Human Feedback (RLHF) has emerged to be one of the important techniques for aligning large language models (LLMs) to human intentions across a diverse set of tasks. Successful applications of RLHF range from chatbots (Ouyang et al., 2022; Bai et al., 2022), and robotics (Yang et al., 2024a) to biomolecule engineering (Wang et al., 2024). Existing RLHF methods work by collecting preference dataset on a given task, and updating a base model using preference based reinforcement learning. Moreover, the availability of many public preference datasets (Shin et al., 2023) has led to the rapid adoption of RLHF across various domains.

However, real-world deployment of fine-tuned policies faces several challenges. Prior work (Stiennon et al., 2020; Gao et al., 2023; Rafailov et al., 2024a) has observed the *overoptimization* phenomenon, where policy optimization improves proxy reward but worsens the true reward function. Moreover, models don't generalize beyond the class of preference examples provided during fine-tuning, and this problem becomes severe when the downstream task undergoes *distribution shift*, e.g. see (Tien et al., 2022). For example, in robotics, the environment encountered during inference might be different than the settings used during training. It is also possible that human preferences drift over time (Son et al., 2024), causing a natural distribution shift.

These examples suggest that in order to truly align LLMs to human intentions, we want a model (reward, policy or both) that generalizes to unseen tasks. In other words, the performance of the fine-tuned model should not deteriorate when the task at inference time differs from the datasets used in fine-tuning. Prior work has mainly considered shifts in preference model (Chowdhury et al., 2024), strong adversarial data corruption model (Mandal et al., 2024), or proposed heuristics like *model merging* (Jang et al., 2023; Ramé et al., 2025) that fails with mild distributional shifts. In this work, we tackle this problem through the lens of *distributionally robust optimization* (DRO) (Ben-Tal et al., 2013) where there is a rich literature on both provable and empirically successful robust algorithms.

Additionally, we also recognize that the existing RLHF pipeline is very efficient because of the development of libraries, sharing of models and datasets on platforms like

Hugging Face. In particular, this has led to the rapid development and adoption of RLHF methods across diverse domains. Therefore, an ideal robust RLHF method should minimally alter existing algorithms. This leads to the two main questions we study in the paper.

> *Can we design provable algorithms for **distributionally robust RLHF**? Can we achieve **OOD robustness** with simple modifications to existing pipeline?*

### 1.1. Our Contributions

We develop distributionally robust algorithms for RLHF by robustifying both the *reward estimation* and the *policy optimization* phase. Our algorithms are based on minibatch gradient descent, easy to implement, and when implemented on base models with 2B parameters (`Gemma-2B-it`) and 7B parameters (`Mistral-7B-Instruct`), shows OOD robustness on various unseen datasets. Our specific contributions are the following.

- **Distributionally Robust RLHF**: We first propose a distributionally robust version of RLHF based on minibatch gradient descent (Levy et al., 2020), and prove a bound on its sample and iteration complexity. In paricular, we show that, under the *linear reward* model, the robust reward estimation phase requires $T = O(1/\varepsilon^2)$ iterations and $n = \widetilde{O}(1/\varepsilon^2)$ samples per iteration.
- **Robust Natural Policy Gradient**: For the robust policy optimization, we propose a new algorithm based on a re-weighted version of natural policy gradient (NPG) method. Under the assumpton of *log-linear policy* class, and additional assumptions, we show that the proposed method has linear convergence rate (i.e. $T = \log(1/\varepsilon)$), but requires $n = \widetilde{O}(1/\varepsilon^4)$ samples per iteration.
- **Robust Direct Preference Optimization**: We also propose robust version of *direct preference optimization* (DPO) (Rafailov et al., 2024b) and analyze its iteration and sample complexity. We show that its iteration ($T = O(1/\varepsilon^2)$) and sample complexity ($n = \widetilde{O}(1/\varepsilon^2)$) are similar to the robust reward estimation step of RLHF.
- **Evaluation of OOD Performance**: We evaluate our robust methods by training base models with 2B and 7B parameters on the Unified-Feedback dataset, and evaluating its performance on out-of-distribution (OOD) datasets e.g. HHH-Alignment, and MT Bench. Across all three methods, we observe that the distributionally robust training provides improved performance on OOD tasks. Furthermore, we observe marked improvement on the reasoning task, and provide fine-grained evaluation on such subsets.

### 1.2. Related Work

The closest to our work is the work by Fisch et al. (2024), who consider robust policy optimization over a class of target reward models for RLHF. However, it is unclear how to choose such a target class of reward models. Similarly, Yan et al. (2024) consider Bayesian Ensembles to model the uncertainty set of reward functions. Concurrent to our work, Xu et al. (2025) has considered distributionally-robust version of DPO, and proposed tractable algorithms under Wasserstein and KL-uncertainty sets. Compared to their setting, we consider TV-distance based uncertainty sets, and consider distributionally robust version of both DPO and RLHF (i.e. reward estimation and policy optimization).

Prior work has also considered other types of robustness in RLHF. Hong et al. (2024) consider robustness in preference model, and assumes a specific type of perturbation ($\mathcal{P}(y^+ \succ y^-|x) = \sigma((r(x, y^+) - r(x, y^-))/\tau)$). Similarly, (Wu et al., 2024a) study distributionally robust version of DPO, but they only consider robustness to the presence of erroneous data pairs in preference rankings. Chowdhury et al. (2024) assumes that the labels are flipped with probability $\varepsilon$ and derives a robust version of direct preference optimization (DPO) to handle this situation. Mandal et al. (2024) assumes that an $\varepsilon$-fraction of the offline preference dataset is corrupted and considers a data corruption robust variant of RLHF. Bukharin et al. (2025) also consider the design of RLHF under corrupted human feedback, however they model corruption as a perturbation of the Bradley-Terry model that generates the preferences. Finally, Ramesh et al. (2024) has adopted the notion of group robustness in RLHF.

Model averaging is a popular approach to improve robustness in RL. Ramé et al. (2025) train $M$ different reward models starting from the same initial pre-trained model, and then take average of them. The authors found that the averaged model performs better on out-of-distribution (OOD) data compared to prediction ensembles (Coste et al., 2023). Yang et al. (2024b) use a shared representation for the policy and the reward model, and then jointly train policy and reward heads to improve robustness on OOD data. Our work is also related to uncertainty-aware RLHF (Lou et al., 2024; Banerjee & Gopalan, 2024), however, these approaches attempt to capture uncertainty and variability in human preferences, and are related to the overoptimization problem in reward estimation (Zhang et al., 2024; Rafailov et al., 2024a).

Finally, we adopt distributionally robust optimization (DRO) (Ben-Tal et al., 2013; Shapiro, 2017), and in particular, adopt the minibatch gradient descent based approach proposed by Levy et al. (2020). A host of methods have been proposed for DRO with various types of shifts in distributions (Mohajerin Esfahani & Kuhn, 2018; Sinha et al., 2017; Staib & Jegelka, 2019) and we refer the interested

reader to the survey (Rahimian & Mehrotra, 2019). We also analyze a DRO version of natural policy gradient that might be of interest beyond RLHF.

Beyond distributionally robust optimization, data selection and mixing have been used as effective strategies to improve robustness of LLMs. Albalak et al. (2023) has used multi-armed bandits algorithms to devise an online data mixing strategy for pre-training LLMs. Recently, Pan et al. (2024) has used bilevel optimization for reweighting datasets during training large-scale LLMs. Xia et al. (2023) introduces dynamic batch loading which adjusts the composition of the sampled batch based on the training loss, an approach very similar to the (reweighted) minibatch gradient descent considered in this paper. Besides pre-training, Corrado et al. (2025) considers direct preference optimization, and adaptively mixes datasets during training to balance performance across tasks. Although data mixing and selection improves LLM training, they have been primarily used in the pre-training stage. Additionally, they lack the principled approach provided by DRO regarding the distance to the target distribution.

## 2. Preliminaries

RLHF, as proposed by Christiano et al. (2017), and later adapted by Ouyang et al. (2022) first learns a reward model from preference data, and then finds a policy that maximizes KL-regularized value function. In particular, we assume access to a preference dataset $\mathcal{D}_{\text{src}} = \{(x, y^+, y^-)\}$ where $x$ is a prompt and $y^+, y^-$ are two possible completions of the prompt $x$ generated from $\pi_{\text{src}}(\cdot \mid x)$ and $\pi_{\text{src}}$ is a reference policy e.g. a supervised fine-tuned policy. Then a preference feedback $y^+ \succ y^- | x$ is labeled by some expert annotator. We assume Bradley-Terry (BT) model which posits

$$
\begin{aligned}
\mathcal{P}\left(y^+ \succ y^- | x\right) &= \sigma(r^\star(x, y^+) - r^\star(x, y^-)) \\
&= 1/(1 + \exp(r^\star(x, y^+) - r^\star(x, y^-)))
\end{aligned}
$$

Here $r^\star(x, y)$ is an underlying reward function of the expert annotator which is the reward assigned to a completion $y$ to a given prompt $x$. The first step of RLHF involves solving a maximum likelihood estimation problem to estimate the reward model.

$$
\widehat{r} \leftarrow \arg\min_r \; -\mathbb{E}_{(x, y^+, y^-) \sim \mathcal{D}_{\text{src}}} \left[\log \sigma\left(r(x, y^+) - r(x, y^-)\right)\right]
$$

The second step of RLHF involves a KL-regularized policy optimization step.

$$
\widehat{\pi} \leftarrow \arg\max_\pi \; \mathbb{E}_{\substack{x \sim \mathcal{D}_{\text{src}} \\ y \sim \pi(\cdot | x)}} \left[\widehat{r}(x, y) - \beta \log \frac{\pi(y|x)}{\pi_{\text{ref}}(y|x)}\right]
$$

Here $\pi_{\text{ref}}$ is a reference policy which can be different than the policy $\pi_{\text{src}}$ used to generate the preference dataset. We will

also write $Y$ to denote the set of all possible completions.

Another popular approach to fine-tuning is the direct preference optimization (DPO) (Rafailov et al., 2024b) which avoids learning an explicit reward model. DPO solves the following optimization problem.

$$
\begin{aligned}
\max_\pi \; \mathbb{E}_{(x, y^+, y^-) \sim \mathcal{D}} \Bigg[ &- \log \sigma \Bigg( \beta \log \frac{\pi(y^+|x)}{\pi_{\text{ref}}(y^+|x)} \\
&- \beta \log \frac{\pi(y^-|x)}{\pi_{\text{ref}}(y^-|x)} \Bigg) \Bigg]
\end{aligned}
$$

## 3. Robust RLHF

We propose a distributionally robust version of RLHF by robustifying the two steps – *reward estimation* and *policy optimization*. In the first step, we learn a robust reward model that considers all distributions that are at a distance of $\rho$ from the training distribution.

$$
\min_r \; \max_{\mathcal{D}: d_\phi(\mathcal{D}, \mathcal{D}_{\text{src}}) \leq \rho} \; -\mathbb{E}_{(x, y^+, y^-) \sim \mathcal{D}} \left[\log \sigma(r(x, y^+) - r(x, y^-))\right]
\tag{1}
$$

Let $\widehat{r}$ be the robust reward model. Then in the second step, we solve a robust version of the KL-regularized policy optimization step.

$$
\max_\pi \; \min_{\mathcal{D}: d_\phi(\mathcal{D}, \mathcal{D}_{\text{src}}) \leq \rho} \; \mathbb{E}_{\substack{x \sim \mathcal{D} \\ y \sim \pi(\cdot|x)}} \left[\widehat{r}(x, y) - \beta \log \frac{\pi(x, y)}{\pi_{\text{ref}}(x, y)}\right]
\tag{2}
$$

Here $d_\phi(Q, P)$ is the $\phi$-divergence between two distributions $Q$ and $P$ defined as $d_\phi(Q, P) := \int \phi\left(\frac{dQ}{dP}\right) dP$ which satisfies $d_\phi(Q, P) \geq 0$ and $d_\phi(Q, P) = 0$ if $Q = P$ a.s. Some common examples of $\phi$ divergences are – (1) $\phi(t) = \frac{1}{2}|t - 1|$ which corresponds to the Total Variation (TV) distance, and (2) $\phi(t) = \frac{1}{2}(t - 1)^2$ which corresponds to the $\chi^2$-distance. In this work, we mainly consider the TV distance.[1]

Algorithm 1 shows a minibatch gradient descent based algorithm to solve the distributionally robust reward estimation. We sample a minibatch $B_t$ of size $n$ (line 3) and compute the optimal weights $q_t^\star$ corresponding to the worst distribution shift within distance $\rho$. Then the minibatch gradient is reweighted by the weights $q_t^\star$ and a gradient descent step is performed for the reward parameters $\theta_t$. Note that, in line 8, we compute an average of the $T$ reward parameters $\{\theta_t\}_{t=1}^T$. This is the correct approach when the loss function is convex e.g. when the reward model is linear. For general reward models, we keep the best model encountered over the $T$ iterations.

Algorithm 2 performs robust policy optimization. This is similar to the minibatch gradient descent steps of the ro-

---

[1]The appendix contains experimental results regarding the $\chi^2$-divergence.

**Algorithm 1** Robust Reward Estimation

**Require:** Preference Dataset $\mathcal{D} = \{(x_i, y_i^+, y_i^-)\}_{i=1}^N$, number of iterations $T$, and stepsize $\eta$.

1: Initialize $r = r_{\omega_1}$ for some $\omega_1 \in \mathcal{W}$.
2: **for** $t = 1, 2, \ldots, T$ **do**
3:    Sample a minibatch $B_t$ of size $n$ from $\mathcal{D}$.
    ▷ Compute optimal weights for the minibatch $B_t$.
4:    Set $q_t^\star$ as the solution of the following problem.

$$\max_{q \in \Delta^b : d_\phi(q, 1/n) \leq \rho} - \sum_{i \in B_t} q_i \cdot \log \sigma\left(r_{\omega_t}(x_i, y_i^+)\right. \\ \left. - r_{\omega_t}(x_i, y_i^-)\right)$$

5:    Set gradient as

$$g_t = \sum_{i \in B_t} q_{t,i}^\star \cdot \nabla_{\omega_t} -\log \sigma\left(r_{\omega_t}(x_i, y_i^+) - r_{\omega_t}(x_i, y_i^-)\right)$$

6:    $\omega_{t+1} = \Pi_{\mathcal{W}}(\omega_t - \eta \cdot g_t)$
7: **end for**
8: $\widehat{r} = r_{\overline{\omega}}$ where $\overline{\omega} = \frac{1}{T}\sum_{t=1}^T \omega_t$.
9: **return** $\widehat{r}$.

---

**Algorithm 2** Robust Policy Optimization

**Require:** Preference Dataset $\mathcal{D} = \{(x_i, y_i^+, y_i^-)\}_{i=1}^N$, number of iterations $T$, estimated reward model $\widehat{r}$, and stepsize $\eta$.

1: Initialize $\pi$ so that $\pi(y|x) = 1/|Y|$ for any $x, y$.
2: **for** $t = 1, 2, \ldots, T$ **do**
3:    Sample a minibatch $B_t$ of size $n$ from $\mathcal{D}$.
4:    Let $\widetilde{B}_t = \{(x_i, y_i) : x_i \in B_t, y_i \sim \pi_{\theta_t}(\cdot|x_i)\}$
    ▷ Compute optimal weights for the minibatch $\widetilde{B}_t$.
5:    Set $q_t^\star \in \arg\min_{q \in \Delta^n : d_\phi(q, 1/n) \leq \rho} \sum_{i \in \widetilde{B}_t} q_i \cdot \left(\widehat{r}(x_i, y_i) - \beta \log \frac{\pi_{\theta_t}(y_i|x_i)}{\pi_{\mathrm{ref}}(y_i|x_i)}\right)$
6:    Set gradient $g_t$ as $\sum_{i \in \widetilde{B}_t} q_{t,i}^\star \cdot \nabla_{\theta_t} \mathbb{E}_{y \sim \pi_{\theta_t}(\cdot|x_i)}\left[\widehat{r}(x_i, y) - \beta \log \frac{\pi_{\theta_t}(y|x_i)}{\pi_{\mathrm{ref}}(y|x_i)}\right]$
    ▷ Compute weighted Fisher information matrix
7:    Let $G^{\pi_{\theta_t}}$ be the matrix

$$\sum_{i=1}^n q_{t,i}^\star \cdot \mathbb{E}_{y \sim \pi_{\theta_t}(\cdot|x_i)}\left[\nabla_{\theta_t} \log \pi_{\theta_t}(y|x_i)\nabla_{\theta_t} \log \pi_{\theta_t}(y|x_i)^\top\right]$$

8:    $\theta_{t+1} = \Pi_\Theta\left(\theta_t + \eta \cdot (G^{\pi_{\theta_t}})^\dagger g_t\right)$
9: **end for**
10: **return** policy $\pi_T = \pi_{\theta_T}$.

---

bust reward estimation, except that the loss function is KL-regularized value function.

$$-\mathbb{E}_{y \sim \pi_\theta(\cdot|x_i)}\left[\widehat{r}(x_i, y) - \beta \log \frac{\pi_\theta(y|x_i)}{\pi_{\mathrm{ref}}(y|x_i)}\right]$$

To evaluate the loss function, we need completions generated from the policy $\pi_{\theta_t}$. Hence, we update the current minibatch $B_t$ to the minibatch $\widetilde{B}_t$ by adding completions $y_i$ generated from $\pi_{\theta_t}(\cdot|x_i)$. Then we compute the optimal weights for the minibatch $\widetilde{B}_t$ and perform a *natural policy gradient ascent* step based on the weighted gradient $g_t$ and the weighted Fisher information matrix $G^{\pi_{\theta_t}}$.

The standard natural policy gradient (Kakade, 2001) first computes the Fisher information matrix $F(\theta) = \mathbb{E}_{x, y \sim \pi_\theta(\cdot|x)}[\nabla_\theta \log \pi_\theta(y|x)\nabla_\theta \log \pi_\theta(y|x)^\top]$ and then performs the update $\theta_{t+1} = \theta_t + \eta(F(\theta_t))^\dagger \nabla_\theta V(\theta_t)$ where $V(\theta_t)$ is the value function of the policy $\pi_{\theta_t}$. Since our goal is to optimize the robust value function, we first compute the weighted Fisher information matrix $G^{\pi_{\theta_t}}$ using the minibatch $\widetilde{B}_t$ (line 7).

$$G^{\pi_{\theta_t}} = \sum_{i=1}^n q_{t,i}^\star \cdot \mathbb{E}_{y \sim \pi_{\theta_t}(\cdot|x_i)}\left[\nabla_{\theta_t} \log \pi_{\theta_t}(y|x_i)\nabla_{\theta_t} \log \pi_{\theta_t}(y|x_i)^\top\right]$$

Then we use the weighted gradient $g_t$ to perform the update $\theta_{t+1} = \theta_t + \eta(G^{\pi_{\theta_t}})^\dagger g_t$ (line 8). In the end, policy $\pi_{\theta_T}$ is returned.

### 3.1. Analysis

We assume linear reward function class and log-linear policy class (Nika et al., 2024). We will also assume that $\phi$-divergence is the total variation distance i.e. $d_\phi(Q, P) := \frac{1}{2}\|P - Q\|_1$. Extending our analysis to other distances e.g. $\chi^2$-distance is straightforward and we discuss this at the end of the paper.

#### 3.1.1. ROBUST REWARD ESTIMATION

We first consider the robust reward estimation phase. Let us define the following logistic loss function.

$$\ell_{\mathrm{REW}}(\omega, \mathcal{D}) = -\mathbb{E}_{(x, y^+, y^-) \sim \mathcal{D}}\left[\log \sigma(r_\omega(x, y^+) - r_\omega(x, y^-))\right]$$

Additionally, let $\ell_{\mathrm{REW},\mathrm{TV}}$ be the robust loss function i.e.

$$\ell_{\mathrm{REW},\mathrm{TV}}(\omega, \mathcal{D}_{\mathrm{src}}) = \max_{\mathcal{D} : d_{\mathrm{TV}}(\mathcal{D}, \mathcal{D}_{\mathrm{src}}) \leq \rho} \ell_{\mathrm{REW}}(\omega, \mathcal{D}) \quad (3)$$

In order to show convergence to an optimal solution of the robust loss function, we make the assumption of linear reward function class.

**Assumption 1** (Linear Reward Function Class). *Let $\phi$ be a $d_R$ dimensional feature mapping with $\max_{x,y}\|\phi(x, y)\|_2 \leq 1$ and let $F > 0$. Then*

*the true reward function belongs to the class* $\mathcal{F} = \left\{ r_\omega : r_\omega(x,y) = \phi(x,y)^\top \omega \text{ where } \omega \in \mathbb{R}^{d_R} \text{ and } \|\omega\|_2 \leq F \right\}$

The next theorem provides the convergence guarantees of algorithm 1 under suitable choices of the minibatch size $n$, and number of iterations $T$.

**Theorem 1.** *Suppose assumption 1 holds, Algorithm 1 is run for* $T = O\left(\frac{1}{\varepsilon^2}\right)$ *iterations, and minibatch size* $n$ *satisfies* $\frac{n}{\log n} \geq O\left(\frac{F^2(1+2\rho)^2}{\varepsilon^2}\right)$ . *Then the solution* $\overline{\omega} = \frac{1}{T} \sum_{t=1}^T \omega_t$ *satisfies*

$$\mathbb{E}\left[\ell_{\text{REW,TV}}(\overline{\omega}, \mathcal{D}_{\text{src}})\right] - \min_\omega \ell_{\text{REW,TV}}(\omega, \mathcal{D}_{\text{src}}) \leq O(\varepsilon).$$

The proof of the Theorem is provided in appendix A.1. The main idea is to bound the bias of the robust loss (as defined in eq. (3)) when estimated on a minibatch of size $n$. In particular, let us define

$$\overline{\ell}_{\text{REW,TV}}(\theta; n) = \mathbb{E}_{x^n} \left[ \sup_{\substack{q \in \Delta^n: \\ d_{\text{TV}}(q, 1/n) \leq \rho}} \sum_{i=1}^n q_i \ell_{\text{REW}}(\theta; x_i) \right].$$

Then using a dual characterization developed by (Levy et al., 2020), we show that the bias term, i.e. the difference between $\ell_{\text{REW,TV}}(\theta; \mathcal{D}_{\text{src}})$ and $\overline{\ell}_{\text{REW,TV}}(\theta; n)$ can be bounded by $O\left(\sqrt{\log n/n}\right)$. [2] The proof then follows by observing that algorithm 1 performs stochastic gradient descent over the loss function $\overline{\ell}_{\text{REW,TV}}(\theta; n)$ which is convex under the assumption of linear reward function class.

### 3.1.2. ROBUST POLICY OPTIMIZATION

We now focus on the performance of robust policy optimization. We will assume that the class of policies is log-linear.

**Assumption 2** (Log-Linear Policy Class). *Let* $\psi$ *be a* $d_P$ *dimensional feature mapping with* $\max_{x,y} \|\psi(x,y)\|_2 \leq 1$ *and let* $B > 0$. *We consider the following class of policies:*

$$\Pi = \left\{ \pi_\theta : \pi_\theta(y|x) = \frac{\exp(\theta^\top \psi(x,y))}{\sum_{y'} \exp(\theta^\top \psi(x,y'))} \right.$$
$$\left. \text{where } \theta \in \mathbb{R}^{d_P} \text{ and } \|\theta\|_2 \leq B \right\}$$

Let us define the following notion of KL-regularized value function.

$$v(\theta; \mathcal{D}) = \mathbb{E}_{x \sim \mathcal{D}, y \sim \pi_\theta(\cdot|x)} \left[ r(x,y) - \beta \cdot \log \frac{\pi_\theta(y|x)}{\pi_{\text{ref}}(y|x)} \right]$$

Additionally, let $v_{\text{TV}}(\theta; \mathcal{D})$ be the robust value function.

$$v_{\text{TV}}(\theta; \mathcal{D}_{\text{src}}) = \inf_{\mathcal{D}: d_{\text{TV}}(\mathcal{D}, \mathcal{D}_{\text{src}}) \leq \rho} v(\theta; \mathcal{D}) \qquad (4)$$

---

[2] Levy et al. (2020) developed similar bounds for the $\chi^2$-divergence.

In order to the suboptimality of the policy $\pi_{\theta_T}$, we make the following assumptions.

**Assumption 3** (Concentrability). *For each t, there exists a constant* $C_t \leq C$ *such that*

$$\sum_{i=1}^n q_{t,i}^\star \cdot \mathbb{E}_{y \sim \pi_{\theta_t}(\cdot|x_i)} \left[ \left( \frac{\pi_{\theta^\star}(y|x_i)}{\pi_{\theta_t}(y|x_i)} \right)^2 \right] \leq C_t$$

**Assumption 4.** *Let* $F(\theta|x) = \mathbb{E}_{y \sim \pi_\theta(\cdot|x)} \left[ \nabla_\theta \log \pi_\theta(y|x) \nabla_\theta \log \pi_\theta(y|x)^\top \right]$ *be the Fisher information matrix corresponding to the policy* $\pi_\theta$ *and sample x. Then there exists* $\sigma > 0$ *such that for any* $t \geq 1$, $\sigma_{\min} \left( \sum_{i=1}^n q_{t,i}^\star \cdot F(\theta_t|x_i) \right) \geq \sigma$ *with probability at least* $1 - \delta$.

**Assumption 5.** *The rewards are bounded i.e.* $\forall x, y$ $r(x,y) \in [r_{\min}, r_{\max}]$.

Assumption 3 states that the distribution $\pi_{\theta_t}$ overlaps with the optimal policy $\pi_{\theta^\star}$. This assumption is standard in the analysis of natural policy gradient (NPG) ascent (Cayci et al., 2024; Alfano et al., 2023) and is often necessary for linear convergence. Assumption 4 states that the weighted Fisher information matrix is positive-definite. Cayci et al. (2024) has shown that this assumption is satisfied when performing NPG ascent on entropy-regularized value function. We believe this assumption should be satisfied for the robust value function considered in this paper. Next we state the convergence guarantees of Algorithm 2.

**Theorem 2.** *Suppose assumptions 3, 4, 5 hold,* $q_{t,i}^\star \geq q_{\min} \forall t$, *and* $\mathcal{D}_{\text{src}}(x) \geq \nu \forall x$. *If Algorithm 2 is run for* $T \geq O\left( \log\left( \frac{\rho^2/\nu(\beta B^2 + r_{\max})}{\varepsilon} \log |Y| \right) \right)$ *iterations, and the number of minibatch samples* $n \geq O\left( \frac{\sqrt{apx} \cdot \rho^4/\nu^2(\beta B^2 + r_{\max})^6}{\eta \beta^2 q_{\min}^2 \sigma^2} \frac{\log(\log \log(\rho/\nu(\beta B^2 + r_{\max})|Y|/\varepsilon)/\delta)}{\varepsilon^4} \right)$. *Then the policy* $\pi_T = \pi_{\theta_T}$ *satisfies,*

$$\sup_\theta v_{\text{TV}}(\theta; \mathcal{D}_{\text{src}}) - v_{\text{TV}}(\theta_T; \mathcal{D}_{\text{src}}) \leq O(\varepsilon)$$

*with probability at least* $1 - \delta$, *provided* $\eta < \min\left\{ \frac{1}{2\beta q_{\min}}, \frac{q_{\min} r_{\min} \sigma^2}{(6\beta B^2 + r_{\max})^2} \right\}$ *and* $\beta < \frac{r_{\min}}{8B^2}$.

Observe that robust policy optimization has linear convergence rate i.e. the number of iterations required to converge upto an error $\varepsilon$ is $O(\log(1/\varepsilon))$. However, the size of the minibatch $n$ is $O(\log \log \log(1/\varepsilon)/\varepsilon^4)$. Therefore, the total number of required samples is $nT = \widetilde{O}(1/\varepsilon^4)$. This requirement is similar to the result of theorem 1 which showed that the robust reward estimation requires $O(1/\varepsilon^2)$ iterations and $\widetilde{O}(1/\varepsilon^2)$ minibatch samples. However, the main difference with theorem 1 is that we wish to show last-iterate convergence of the robust policy optimization as opposed to the average iterate convergence.

The full proof of Theorem 2 is provided in appendix A.4. The main idea is to define the following potential function, similar to (Cayci et al., 2024).

$$\Phi(\pi) = \sum_x \mathcal{D}_{\text{src}}(x) \sum_y \pi_{\theta^\star}(y|x) \log \frac{\pi_{\theta^\star}(y|x)}{\pi(y|x)}$$

where $\theta^\star$ is the optimal solution to the robust KL-regularized value function in the log-linear policy class. We first establish the following recurrence relation.

$$\Phi(\pi_{t+1}) \le (1 - \eta\beta q_{\min})\Phi(\pi_t) - \frac{\eta}{n}\Delta_t + \frac{\eta\sqrt{\text{apx}}}{n \cdot q_{\min}} + \widetilde{O}\left(\sqrt{\frac{\eta}{n}}\right)$$

where $\text{apx}$ is the function approximation error (defined in appendix A.4), and $\Delta_t$ is the suboptimality gap of the policy $\pi_t = \pi_{\theta_t}$. The recurrence is similar to prior work, except that we have an additive error of $\widetilde{O}(\sqrt{\eta/n})$ because the gradients are computed on a minibatch of size $n$. Equipped with the above recurrence we can show that $\Phi(\pi_T) \le O(\varepsilon)$ if $T \ge O(\log(1/\varepsilon))$ and $n \ge \widetilde{O}(1/\varepsilon^2)$. Furthermore, lemma 3 shows that $\Phi(\pi_\theta) \le \varepsilon$ implies that the suboptimality gap of $\pi_\theta$ is $O(\sqrt{\varepsilon})$. Now, after adjusting the value of $\varepsilon$ we can complete the proof of theorem 2.

## 4. Robust Direct Preference Optimization

We also consider a distributionally robust version of the *direct preference optimization* (DPO) method, where the learner needs to be robust to shifts in prompt distributions.

$$\min_\pi \max_{\mathcal{D}:d_{TV}(\mathcal{D},\mathcal{D}_{\text{src}}) \le \rho} - \mathbb{E}_{(x,y^+,y^-) \sim \mathcal{D}}\left[\log \sigma\left(\beta \log \frac{\pi(y^+|x)}{\pi_{\text{ref}}(y^+|x)}\right.\right.$$
$$\left.\left. - \beta \log \frac{\pi(y^-|x)}{\pi_{\text{ref}}(y^-|x)}\right)\right] \quad (5)$$

Algorithm 3 proposes a minibatch stochastic gradient descent based algorithm for solving the distributionally robust DPO problem. In order to analyze the convergence of algorithm 3, we define the following DPO loss function.

$$\ell_{\text{DPO}}(\theta, \mathcal{D}) = - \mathbb{E}_{(x,y^+,y^-) \sim \mathcal{D}}\left[\log \sigma\left(\beta \log \frac{\pi_\theta(y^+|x)}{\pi_{\text{ref}}(y^+|x)}\right.\right.$$
$$\left.\left. - \beta \log \frac{\pi_\theta(y^-|x)}{\pi_{\text{ref}}(y^-|x)}\right)\right] \quad (6)$$

Additionally, let us write $\ell_{\text{DPO,TV}}(\theta, \mathcal{D})$ to define the robust loss function, i.e.

$$\ell_{\text{DPO,TV}}(\theta, \mathcal{D}_{\text{src}}) = \max_{\mathcal{D}:d_{TV}(\mathcal{D},\mathcal{D}_{\text{src}}) \le \rho} \ell_{\text{DPO}}(\theta, \mathcal{D}). \quad (7)$$

The next theorem provides the convergences guarantees of algorithm 3 under log-linear policy class.

---

**Algorithm 3** Distributionally Robust DPO

**Require:** Preference Dataset $\mathcal{D} = \left\{(x_i, y_i^+, y_i^-)\right\}_{i=1}^N$, and number of iterations $T$.
1: Initialize $\pi = \pi_{\theta_1}$ for some $\theta_1 \in \Theta$, and $\eta = 2/\sqrt{T}$.
2: **for** $t = 1, 2, \ldots, T$ **do**
3:    Sample a minibatch of size $B_t$ of size $n$ from $\mathcal{D}$.
    ▷ Compute optimal weights for the minibatch $B_t$.
4:    Set $q_t^\star \in \arg\max_{q \in \Delta^n : D_{TV}(q,1/n) \le \rho} \sum_{i \in B_t} q_i \cdot$
    $- \log \sigma\left(\beta \log \frac{\pi_{\theta_t}(y^+|x)}{\pi_{\text{ref}}(y^+|x)} - \beta \log \frac{\pi_{\theta_t}(y^-|x)}{\pi_{\text{ref}}(y^-|x)}\right)$
5:    Set gradient $g_t = \sum_{i \in B_t} q_i^\star \cdot \nabla_{\theta_t} -$
    $\log \sigma\left(\beta \log \frac{\pi_{\theta_t}(y^+|x)}{\pi_{\text{ref}}(y^+|x)} - \beta \log \frac{\pi_{\theta_t}(y^-|x)}{\pi_{\text{ref}}(y^-|x)}\right)$
6:    $\theta_{t+1} = \Pi_\Theta(\theta_t - \eta \cdot g_t)$
7: **end for**
8: **return** policy $\pi_{\overline{\theta}}$ where $\overline{\theta} = \frac{1}{T}\sum_{t=1}^T \theta_t$.

---

**Theorem 3.** *Suppose assumption 2 holds, algorithm 3 is run for $T = O\left(\frac{1}{\varepsilon^2}\right)$ iterations, and we set minibatch size $n$ such that $\frac{n}{\log n} \ge O\left(\frac{\beta^2(2B+J)^2(1+2\rho)^2}{\varepsilon^2}\right)$ where $J = \max_{x,y^+,y^-}\left|\log \frac{\pi_{ref}(y^+|x)}{\pi_{ref}(y^-|x)}\right|$. Then we have $\mathbb{E}\left[\ell_{\text{DPO,TV}}(\overline{\theta}, \mathcal{D}_{\text{src}})\right] - \min_\theta \ell_{\text{DPO,TV}}(\theta, \mathcal{D}_{\text{src}}) \le O(\varepsilon)$.*

The proof of this theorem is provided in appendix A.2 and is very similar to the proof of theorem 1. Note that the number of iterations $T$ is $O(1/\varepsilon^2)$ and the number of samples per minibatch is $\widetilde{O}(1/\varepsilon^2)$. Therefore, the total sample complexity of the Distributionally robust DPO is $\widetilde{O}(1/\varepsilon^4)$ which is similar to the Distributionally robust PPO and reward learning phase.

## 5. Experiments

**Setup**. We run our experiments on two open-source models – (a) `google/gemma-2b-it`, a lightweight 2B model released by Google, and (b) `mistralai/Mistral-7B-Instruct-v0.2`, a 7B instruct model released by Mistral. All our models are trained on the Unified-Feedback dataset (Jiang et al., 2023). Our goal is to evaluate the out-of-distribution robustness of our methods, and we leverage RewardBench (Lambert et al., 2024), HHH-Alignment (Askell et al., 2021), and MT-Bench (Zheng et al., 2023).

We rely on the Hugging Face `trl` library (von Werra et al., 2020) to set up our experiments, and update the reward estimation and policy optimization methods in the library. In particular, we compute a re-weighted gradient on minibatches by obtaining the weights $q_t^\star$ corresponding to the worst-case distribution shift (e.g. line 4 of Algorithm 1). This is implemented using `cvxpy` (Dia-

*Table 1.* Evaluation of the reward models on the RewardBench (Lambert et al., 2024) benchmark. Trained on a 400K item subset of the Unified-Feedback dataset (Jiang et al., 2023) for 2 epochs, with the TV distance. Standard errors are derived from five runs. Evaluation using the `mistralai/Mistral-7B-Instruct-v0.2` model can be seen in Appendix, Table 8. Detailed table with subsets can be seen in Appendix, Table 10.

| Reward Model | Chat | Chat-Hard | Safety | Reasoning | Avg. |
|---|---|---|---|---|---|
| Base model: `google/gemma-2b-it` | | | | | |
| BTL RM | $96.5 \pm 0.2$ | $45.0 \pm 0.4$ | $81.0 \pm 0.2$ | $66.8 \pm 1.8$ | $72.3 \pm 0.4$ |
| DR RM $\rho = 0.1$ | $96.6 \pm 0.1$ | $44.3 \pm 0.4$ | $80.8 \pm 0.2$ | $68.9 \pm 0.3$ | $72.7 \pm 0.1$ |
| DR RM $\rho = 0.2$ | $97.1 \pm 0.3$ | $42.5 \pm 0.6$ | $80.9 \pm 0.3$ | $71.4 \pm 1.6$ | $72.9 \pm 0.5$ |
| DR RM $\rho = 0.4$ | $87.2 \pm 2.6$ | $33.5 \pm 3.7$ | $71.2 \pm 2.8$ | $52.1 \pm 7.3$ | $61.0 \pm 4.0$ |
| DR RM $\rho = 0.6$ | $82.3 \pm 4.1$ | $33.6 \pm 4.2$ | $58.2 \pm 5.6$ | $50.5 \pm 5.6$ | $56.2 \pm 4.6$ |
| DR RM $\rho = 0.8$ | $75.4 \pm 6.9$ | $37.3 \pm 4.0$ | $56.5 \pm 5.7$ | $47.4 \pm 7.7$ | $54.1 \pm 5.3$ |
| DR RM $\rho = 1.0$ | $60.6 \pm 12.2$ | $32.3 \pm 7.5$ | $48.2 \pm 9.2$ | $44.7 \pm 10.9$ | $46.4 \pm 8.7$ |

*Table 2.* Evaluation of a single run of the distributionally robust PPO on the RewardBench (Lambert et al., 2024) benchmark. Trained on a 5K item subset of the Unified-Feedback dataset (Jiang et al., 2023) with the TV distance. Indicated $\rho$ used for both reward model training and PPO optimization. Detailed table with subsets can be seen in Appendix, Table 16.

| Model | Chat | Chat-Hard | Safety | Reasoning | Avg. | Reasoning Subsets | | | | | |
|---|---|---|---|---|---|---|---|---|---|---|---|
| BTL RM, Std. PPO | 77.9 | 31.1 | 40.0 | 45.9 | 48.7 | 14.1 | 81.1 | 77.4 | 77.4 | 75.0 | 77.4 77.4 |
| DR PPO $\rho = 0.1$ | 82.1 | 32.0 | 39.3 | 55.9 | 52.3 | 38.3 | 73.8 | 73.2 | 72.0 | 70.7 | 75.0 76.2 |
| DR PPO $\rho = 0.2$ | 79.3 | 32.0 | 40.0 | 48.7 | 50.0 | 17.4 | 81.1 | 81.7 | 81.1 | 78.7 | 80.5 76.8 |
| DR PPO $\rho = 0.4$ | 79.3 | 32.2 | 39.1 | 47.7 | 49.6 | 15.2 | 81.7 | 81.7 | 84.1 | 77.4 | 77.4 79.3 |
| DR PPO $\rho = 0.6$ | 78.5 | 30.3 | 38.5 | 47.9 | 48.8 | 16.1 | 78.7 | 80.5 | 81.1 | 79.9 | 79.3 78.7 |
| DR PPO $\rho = 0.8$ | 78.5 | 29.6 | 40.0 | 50.5 | 49.6 | 32.2 | 67.1 | 63.4 | 72.6 | 67.7 | 70.7 70.7 |
| DR PPO $\rho = 1.0$ | 73.5 | 30.3 | 41.2 | 55.8 | 50.2 | 44.5 | 66.5 | 67.7 | 67.1 | 67.1 | 66.5 68.3 |

mond & Boyd, 2016; Agrawal et al., 2018). We use hyperparameters from a recent work by Yang et al. (2024b), with minimal tuning, the details of which are provided in Appendix B. We present an evaluation of our proposed distributionally robust RLHF and DPO methods, showing the effect it has on in and out-of-distribution performance. For all the experiments, we choose the Total variation (TV) distance as a distance measure between two distributions, and vary the parameter $\rho$ (maximum distance from the training distribution). In the main text we present results with `google/gemma-2b-it`, and the results for `mistralai/Mistral-7B-Instruct-v0.2` are provided in the Appendix. Our code is available in the GitHub repository: `https://github.com/PauliusSasnauskas/dr-rlhf`.

### 5.1. Reward Models

First we evaluate the performance of robust reward estimation. Table 1 shows the performance of the reward models on the RewardBench (Lambert et al., 2024) benchmark. In the tables we denote **BTL RM** – a reward model trained in the standard way (Bradley–Terry loss), and **DR RM** – the distributionally robust version, based on Algorithm 1. We see that as $\rho$ increases the performance of **DR RM** im-

proves up to a certain point, with $\rho = 0.2$ being the optimal choice. For larger $\rho$ the performance drops, since we require the training methods to be robust to a very large class of distribution shifts. We choose TV distance as our distance function. All reward models were trained on a 400K item subset of the Unified-Feedback dataset (Jiang et al., 2023), following the same pipeline as Yang et al. (2024b).

In addition to the overall improvement, we can see that the performance increased significantly in some tasks, such as Reasoning. Therefore, we focus on the Reasoning task, and present more detailed results on these subsets in table Table 3. These results show the method improves performance significantly for certain subsets. We observe an increase in performance for columns 1 (+23.4%), 2 (+3.1%), 5 (+2.4%), and 6 (+5.5%) in the reasoning task.

### 5.2. Policy Optimization

We next consider the robust policy optimization when given an estimated reward model as input. Although algorithm 2 specifies a weighted version of natural policy gradient method, in order to obtain a practical RLHF algorithm, we deviate from NPG and adopt proximal policy optimization (PPO, Schulman et al., 2017), which has been immensely

successful in LLM fine-tuning. However, we implement robust variants of PPO by reweighting the gradients. This was implemented in two ways – (1) scale the rewards/advantages computed by PPO's value function (*version A*), and (2) scale the policy loss (*version B*). Initial experiments showed *version A* is performing slightly better than *version B* (see the comparison in the Appendix, Table 15), and we present only version A in the main text.

*Table 3.* Evaluation of a single run of the distributionally robust reward models on the Reasoning task from the RewardBench (Lambert et al., 2024) benchmark. Trained on a 400K item subset of the Unified-Feedback dataset (Jiang et al., 2023) for 2 epochs, with the TV distance. Subsets are detailed in Appendix B.1.1.

| Reward Model | Reasoning Subsets | | | | | | |
|---|---|---|---|---|---|---|---|
| BTL RM | 47.7 | 82.3 | 83.5 | 86.6 | 79.9 | 81.1 | 83.5 |
| DR RM $\rho = 0.1$ | 55.5 | 78.7 | 83.5 | 83.5 | 78.7 | 84.1 | 79.3 |
| DR RM $\rho = 0.2$ | 61.7 | 83.5 | 83.5 | 85.4 | 81.7 | 86.0 | 83.5 |
| DR RM $\rho = 0.4$ | 71.1 | 84.8 | 81.1 | 83.5 | 79.9 | 81.1 | 82.3 |
| DR RM $\rho = 0.6$ | 55.3 | 81.1 | 81.7 | 83.5 | 79.9 | 86.6 | 83.5 |
| DR RM $\rho = 0.8$ | 66.2 | 85.4 | 78.0 | 81.7 | 82.3 | 84.8 | 82.9 |
| DR RM $\rho = 1.0$ | 70.9 | 81.7 | 78.7 | 82.9 | 80.5 | 85.4 | 81.7 |

In order to train the policy, we use the DR RMs obtained in the previous subsection, i.e. for a given value of $\rho$, we use the robust reward model obtained with the same $\rho$, and then run robust PPO with identical $\rho$. We downsample 5K data from the Unified-Feedback dataset to use as the training data. We use the same choice of $\rho$ as for the reward models. We evaluate our models on the Unified-Feedback test set, HHH Alignment (Askell et al., 2021), and MT-Bench (Zheng et al., 2023) tasks. Table 4 shows the performance of DR PPO, and the results show that for most choices of $\rho$ the performance is comparable or increases for both ID *and* OOD tasks, compared to standard PPO.

*Table 4.* Evaluation of the PPO trained policy on Unified-Feedback (ID, Jiang et al., 2023), HHH Alignment (OOD, Askell et al., 2021), and MT Bench (OOD, Zheng et al., 2023) datasets. Trained on a 5K item subset of the Unified-Feedback dataset. Evaluation using the `mistralai/Mistral-7B-Instruct-v0.2` model can be seen in the Appendix, Table 15.

| Model | Unified-Feedback | HHH Alignment | MT Bench |
|---|---|---|---|
| Base model: `google/gemma-2b-it` | | | |
| Std. PPO | 59.8 ±0.2 | 66.9 ±0.5 | 59.9 ±0.3 |
| DR PPO $\rho = 0.1$ | 59.8 ±0.4 | 68.1 ±0.3 | 61.6 ±1.0 |
| DR PPO $\rho = 0.2$ | 59.6 ±0.2 | 66.2 ±0.7 | 59.5 ±0.4 |
| DR PPO $\rho = 0.4$ | 59.9 ±0.2 | 67.1 ±0.3 | 59.6 ±0.3 |
| DR PPO $\rho = 0.6$ | 60.3 ±0.3 | 67.9 ±0.5 | 59.4 ±0.2 |
| DR PPO $\rho = 0.8$ | 60.2 ±0.3 | 66.7 ±0.7 | 59.6 ±0.4 |
| DR PPO $\rho = 1.0$ | 59.8 ±0.1 | 66.4 ±0.3 | 59.6 ±0.2 |

We additionally run an evaluation of the PPO policy on RewardBench (Lambert et al., 2024). To evaluate PPO on

*Table 5.* Evaluation of the DPO trained policy on Unified-Feedback (ID, Jiang et al., 2023), HHH Alignment (OOD, Askell et al., 2021), and MT Bench (OOD, Zheng et al., 2023) datasets. Trained on a 400K item subset of the Unified-Feedback dataset using TV dist. Evaluation using the `mistralai/Mistral-7B-Instruct-v0.2` model can be seen in the Appendix, Table 17.

| Model | Unified-Feedback | HHH Alignment | MT Bench |
|---|---|---|---|
| Base model: `google/gemma-2b-it` | | | |
| Std. DPO | 60.9±0.02 | 69.2 ±0.1 | 61.1±0.05 |
| DR DPO $\rho = 0.1$ | 61.3±0.02 | 68.9 ±0.2 | 61.9±0.07 |
| DR DPO $\rho = 0.2$ | 61.7±0.02 | 69.0 ±0.0 | 62.5±0.05 |
| DR DPO $\rho = 0.4$ | 62.2 ±0.3 | 68.5 ±0.6 | 64.4 ±0.9 |
| DR DPO $\rho = 0.6$ | 63.8 ±0.2 | 69.3 ±0.3 | 64.1 ±0.8 |
| DR DPO $\rho = 0.8$ | 64.4 ±0.3 | 70.6 ±0.4 | 63.5 ±1.2 |
| DR DPO $\rho = 1.0$ | 64.5 ±0.3 | 64.5 ±2.6 | 62.5 ±0.5 |

RewardBench, we adopt the following convention. For a given prompt $x$, chosen and rejected completions $y^+$ and $y^-$ respectively, the example is correctly classified if $\pi(y^+|x) > \pi(y^-|x)$. The results can be seen in Table 2. Similar to the evaluation of the robust reward model, these results show improvement for some values of $\rho$ (e.g., $\rho \in \{0.1, 0.2, 0.4, 0.8\}$). Furthermore, we again see a sharp improvement in the reasoning task ($+10.0\%$ with $\rho = 0.1$). We analyze the reasoning task in more detail, and again see significant improvement on several subsets e.g., columns $1, 4$, and $5$.

### 5.3. Evaluation of Robust DPO

We base our Robust DPO implementation on Algorithm 3, and train it on the same 400K item subset of the Unified-Feedback dataset as for the reward models. The results for Robust DPO can be seen in Table 5. We see a consistent improvement as $\rho$ increases, for both ID and OOD datasets. Additionally, we see that for OOD datasets, an intermediate value of $\rho$ attains the best performance. For HHH Alignment the corresponding value of $\rho = 0.8$, and for MT Bench it is $\rho = 0.5$. Moreover, the DR DPO performs better than DR PPO (table 4) on OOD datasets. We also evaluate DR DPO on RewardBench, and the details can be found in Appendix, Table 18.

## 6. Conclusion

In this work, we have shown that distributionally robust RLHF algorithms improve OOD robustness of LLMs. Note that, we have primarily presented our results about the Total Variation (TV) distance, however, extension to other distances like $\chi^2$ is also possible if one uses dual characterization from (Levy et al., 2020). We have added additional experiments with $\chi^2$-distance in the Appendix (Table 12),

and and we believe that the result for natural policy gradient (Theorem 2) can be strengthened with the $\chi^2$-divergence.

Throughout our analysis, we have considered linear reward functions. However, Our results for robust reward estimation naturally generalizes for nonlinear and nonconvex reward models. The main observation is that the bias of the minibatch gradient (Lemma 1 in the Appendix) is independent of the choice of the reward model. Since Algorithm 1 performs a biased gradient descent and this implies convergence to a stationary point of the distributionally robust objective. The linearity of the reward function is only used to guarantee convergence to global optima of Algorithm 1. For a general nonconvex reward function, we can easily establish convergence to a stationary point and, with additional structural assumptions (e.g. standard Polyak-Łojasiewicz condition (Karimi et al., 2016)) we can obtain convergence to global optima.

We have developed robust variants of popular policy optimization methods under the assumption that human preferences are transitive. There is a host of methods that are designed to solve preference alignment with generalized non-transitive preferences like Nash-MD (Munos et al., 2023), Self Play Preference Optimization (SPPO) (Wu et al., 2024b), etc. It will be interesting to develop distributionally robust variants of these methods.

It will also be interesting to consider distance metrics that are designed specifically for text domains e.g., Zeng et al. (2024). We believe that our techniques will still be useful for different metrics, however, the interesting question is devising such a metric. Finally, we are really excited to see that the DRO version of RLHF improves reasoning capabilities significantly on OOD datasets. It is not a prior clear why enforcing DRO should positively affect the performance on reasoning tasks, and it will be worth investigating this phenomenon further and understand if there are other connections between reasoning and distributional robustness.

## Acknowledgements

The work of Goran Radanovic was funded by the Deutsche Forschungsgemeinschaft (DFG, German Research Foundation) – project number 467367360. We acknowledge the use of computing resources generously provided by MPI-SWS.

## Impact Statement

We expect our distributionally robust RLHF methods to greatly improve the robustness of LLMs. The methods are applicable whenever there is a shift in the prompt distribution during deployment. In particular, we anticipate the distributionally robust methods to be useful in applications of LLMs in healthcare. They might also be useful in multilingual LLMs, particularly in answering prompts from low-resource languages. This paper presents work whose goal is to advance the field of reinforcement learning from human feedback. We don't anticipate any negative consequences of our work.

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

# Appendix

## A. Missing Proofs

Let $x^n \sim P^n$ be a sample of size of $n$ drawn from the distribution $P$. Given a loss function $\ell$ and divergence function $\phi$, let us define the following loss functions.

$$\ell_\phi(\theta; x^n) = \sup_{q \in \Delta^n : \frac{1}{n} \sum_{i=1}^n \phi(nq_i) \le \rho} \sum_{i=1}^n q_i \ell(\theta; x_i)$$

$$\bar{\ell}_\phi(\theta; n) = \mathbb{E}_{x^n \sim P^n} [\ell(\theta; x^n)]$$

We first provide a bound on the bias $\ell_\phi(\theta; P_0) - \bar{\ell}_\phi(\theta; n)$ for the divergence function $\phi(t) = \frac{1}{2} |t - 1|$, which corresponds to the Total variation distance. With slight abuse of notation, we will write the corresponding loss function as $\ell_{TV}(\theta; \cdot)$.

**Lemma 1.** *Suppose $|\ell(\theta; x)| \le B$ for any $\theta$ and $x$. Then we have*

$$0 \le \ell_{\text{TV}}(\theta; P) - \bar{\ell}_{\text{TV}}(\theta; n) \le 3B(1 + 2\rho) \cdot \sqrt{\frac{4 + \log n}{n}}.$$

*Proof.* Let $F^{-1}(u)$ be the inverse cumulative distribution function of the loss function $\ell(\theta; x)$ under the distribution $P$. To be precise define $F(t) = \mathbb{P}_{x \sim P}(\ell(\theta; x) \le t)$, and $F^{-1}(u) = \inf \{t : F(t) \ge u\}$. Then we can obtain the inverse-cdf formulation from (Levy et al., 2020) (eq. 20) to obtain the following result.

$$\ell_{\text{TV}}(\theta; P) = \sup_{r \in \mathcal{R}} \int_0^1 r(u) F^{-1}(u) du$$

$$\text{where } \mathcal{R} = \left\{ r : [0, 1] \to \mathbb{R}_+ | \int_0^1 r(u) du = 1, \int_0^1 \frac{1}{2} |r(u) - 1| \, du \le \rho \text{ and } r \text{ non-increasing} \right\}$$

Now we can proceed similar to the proof of bound (9) from (Levy et al., 2020) and establish that $\ell_{\text{TV}}(\theta; P) \le \bar{\ell}_{\text{TV}}(\theta; n) + E$ with

$$E = \int_0^1 [r(\alpha) - r(1)] (\alpha \cdot bb(\alpha))' \, d\alpha \quad \text{where } bb(\alpha) = 3B \cdot \min \left\{ 1, \sqrt{\frac{1}{\alpha n}} \right\}$$

We can bound the error term $E$ using the Cauchy Schwarz inequality.

$$E \le \|r\|_2 \cdot \left\| (\alpha \cdot bb(\alpha))' \right\|_2$$

Now for any $r \in \mathcal{R}$, we can bound the first term as $\|r\|_2 \le \|r\|_1 \le \int_0^1 |r(u) - 1| \, du + 1 \le 1 + 2\rho$. The proof of bound (9) from (Levy et al., 2020) bound the term $\left\| (\alpha \cdot bb(\alpha))' \right\|_2$ by $3B \cdot \sqrt{\frac{4 + \log n}{n}}$. $\qquad \square$

### A.1. Proof of theorem 1

*Proof.* From the assumption of linear reward class (assumption 1) and bounded features, we have

$$\left| \ell_{\text{REW}}(\omega; x, y^+, y^-) \right| = \left| -\log \sigma \left( r_\omega(x, y^+) - r_\omega(x, y^-) \right) \right|$$
$$= \left| \log \sigma \left( \omega^\top (\psi(x, y^+) - \psi(x, y^-)) \right) \right| \le 4F$$

The last inequality follows from the following observations. First, $\omega^\top (\psi(x, y^+) - \psi(x, y^-)) \le \|\omega\|_2 (\|\psi(x, y^+)\|_2 + \|\psi(x, y^-)\|_2) \le 2F$. This implies that the term inside the sigmoid function is bounded between $-2F$ and $2F$, and for any $u$ we have $|\log \sigma(u)| \le 2|u|$.

We can now apply lemma 1 to the above loss function and bound the bias of the minibatch sample as follows.

$$\ell_{\mathrm{REW,TV}}(\omega; \mathcal{D}_{\mathrm{src}}) - \bar{\ell}_{\mathrm{REW,TV}}(\omega; n) \leq \underbrace{12F(1+2\rho) \cdot \sqrt{\frac{4+\log n}{n}}}_{:=\delta}$$

algorithm 1 performs a stochastic gradient descent on the loss function $\bar{\ell}_{\mathrm{REW,TV}}(\omega; n)$, which is $\delta$-biased with respect to the loss function $\ell_{\mathrm{REW,TV}}(\omega; \mathcal{D})$. Therefore, we can apply convergence guarantees for the (biased) stochastic gradient method as long as we have a bound on the norm of the subgradient. For the linear reward model (assumption 1) we have,

$$\begin{aligned}
\nabla_\omega \ell_{\mathrm{REW}}(\omega; x, y^+, y^-) &= -\frac{\sigma'\left(\omega^\top\left(\psi(x,y^+) - \psi(x,y^-)\right)\right)}{\sigma\left(\omega^\top\left(\psi(x,y^+) - \psi(x,y^-)\right)\right)} \cdot \left(\psi(x,y^+) - \psi(x,y^-)\right) \\
&= -\left(1 - \sigma\left(\omega^\top\left(\psi(x,y^+) - \psi(x,y^-)\right)\right)\right)\left(\psi(x,y^+) - \psi(x,y^-)\right)
\end{aligned}$$

using the fact $\sigma'(u) = \sigma(u)(1-\sigma(u))$. This implies $\|\nabla_\omega \ell_{\mathrm{REW}}(\omega; x, y^+, y^-)\|_2 \leq \|\psi(x,y^+)\|_2 + \|\psi(x,y^-)\|_2 \leq 2$. Now for a minibatch $B$ of size $n$, the gradient of the loss function $\ell_{\mathrm{REW,TV}}(\omega; n)$ can be expressed as $\widetilde{g} = \sum_{i=1}^n q_i^\star \cdot \nabla_\omega \ell_{\mathrm{REW}}(\omega; x_i, y_i^+, y_i^-)$ where $\{q_i^\star\}_{i=1}^n$ is the solution to the optimization problem $\max_{q \in \Delta^n, D_{\mathrm{TV}}(q, 1/n) \leq \rho} \sum_{i=1}^n q_i \cdot \ell_{\mathrm{REW}}(\omega; x_i, y_i^+, y_i^-)$. This implies $\|\widetilde{g}\|_2 \leq 2 \sum_{i=1}^n q_i^\star = 2$. Therefore, we can apply proposition 1 to obtain the following bound.

$$\mathbb{E}\left[\ell_{\mathrm{REW,TV}}(\overline{\omega}; \mathcal{D}_{\mathrm{src}})\right] - \min_\omega \ell_{\mathrm{REW,TV}}(\omega; \mathcal{D}_{\mathrm{src}}) \leq \delta + \frac{2}{\sqrt{T}}$$

$$\leq 12F(1+2\rho) \cdot \sqrt{\frac{4+\log n}{n}} + \frac{2}{\sqrt{T}}$$

Now if we set $T = O(1/\varepsilon^2)$ and choose minibatch size so that $n/\log n = O\left(\frac{B^2(1+2\rho)^2}{\varepsilon^2}\right)$, we obtain the upper bound of $O(\varepsilon)$. $\qquad\square$

## A.2. Proof of theorem 3

*Proof.* From the assumption of log-linear policy class (assumption 2) and bounded features, we have

$$\begin{aligned}
\left|\ell_{\mathrm{DPO}}(\theta; x, y^+, y^-)\right| &= \left|-\log \sigma\left(\beta \log \frac{\pi_\theta(y^+|x)}{\pi_{\mathrm{ref}}(y^+|x)} - \beta \log \frac{\pi_\theta(y^-|x)}{\pi_{\mathrm{ref}}(y^-|x)}\right)\right| \\
&= \left|\log \sigma\left(\beta \log \frac{\pi_\theta(y^+|x)}{\pi_\theta(y^-|x)} \cdot \frac{\pi_{\mathrm{ref}}(y^+|x)}{\pi_{\mathrm{ref}}(y^-|x)}\right)\right| \\
&= \left|\log \sigma\left(\beta \log \frac{\exp(\theta^\top \psi(x,y^+))}{\exp(\theta^\top \psi(x,y^-))} \cdot \frac{\pi_{\mathrm{ref}}(y^+|x)}{\pi_{\mathrm{ref}}(y^-|x)}\right)\right| \\
&= \left|\log \sigma\left(\beta \theta^\top\left(\psi(x,y^+) - \psi(x,y^-)\right) + \beta \log \frac{\pi_{\mathrm{ref}}(y^+|x)}{\pi_{\mathrm{ref}}(y^-|x)}\right)\right| \leq 2\beta(2B+J)
\end{aligned}$$

The last inequality follows from the following observations. First, $\beta\theta^\top(\psi(x,y^+) - \psi(x,y^-)) \leq \beta\|\theta\|_2(\|\psi(x,y^+)\|_2 + \|\psi(x,y^-)\|_2) \leq 2\beta B$. This implies that the term inside the sigmoid function is bounded between $-\beta(2B+J)$ and $\beta(2B+J)$, and for any $u$ we have $|\log \sigma(u)| \leq 2|u|$.

We can now apply lemma 1 to the DPO loss function and bound the bias of the minibatch sample as follows.

$$\ell_{\mathrm{DPO,TV}}(\theta; \mathcal{D}_{\mathrm{src}}) - \bar{\ell}_{\mathrm{DPO,TV}}(\theta; n) \leq \underbrace{6\beta(2B+J)(1+2\rho) \cdot \sqrt{\frac{4+\log n}{n}}}_{:=\delta}$$

algorithm 3 performs a stochastic gradient descent on the loss function $\bar{\ell}_{\mathrm{DPO,TV}}(\theta; n)$, which is $\delta$-biased with respect to the loss function $\ell_{\mathrm{DPO,TV}}(\theta; \mathcal{D})$. Therefore, we can apply convergence guarantees for the (biased) stochastic gradient method

as long as we have a bound on the norm of the subgradient. For the log-linear policy class (assumption 2) we have,

$$\nabla_\theta \ell_{\text{DPO}}(\theta; x, y^+, y^-) = -\frac{\sigma'\left(\beta\theta^\top(\psi(x, y^+) - \psi(x, y^-)) + \beta\log\frac{\pi_{\text{ref}}(y^+|x)}{\pi_{\text{ref}}(y^-|x)}\right)}{\sigma\left(\beta\theta^\top(\psi(x, y^+) - \psi(x, y^-)) + \beta\log\frac{\pi_{\text{ref}}(y^+|x)}{\pi_{\text{ref}}(y^-|x)}\right)} \cdot (\psi(x, y^+) - \psi(x, y^-))$$

$$= -\left(1 - \sigma\left(\beta\theta^\top(\psi(x, y^+) - \psi(x, y^-)) + \beta\log\frac{\pi_{\text{ref}}(y^+|x)}{\pi_{\text{ref}}(y^-|x)}\right)\right)(\psi(x, y^+) - \psi(x, y^-))$$

using the fact $\sigma'(u) = \sigma(u)(1 - \sigma(u))$. This implies $\|\nabla_\theta\ell_{\text{DPO}}(\theta; x, y^+, y^-)\|_2 \leq \|\psi(x, y^+)\|_2 + \|\psi(x, y^-)\|_2 \leq 2$. Now for a minibatch $B$ of size $n$, the gradient of the loss function $\ell_{\text{DPO,TV}}(\theta; n)$ can be expressed as $\widetilde{g} = \sum_{i=1}^n q_i^\star \cdot \nabla_\theta\ell_{\text{DPO}}(\theta; x_i, y_i^+, y_i^-)$ where $\{q_i^\star\}_{i=1}^n$ is the solution to the optimization problem $\max_{q\in\Delta^n, D_{\text{TV}}(q, 1/n)\leq\rho} \sum_{i=1}^n q_i \cdot \ell_{\text{DPO}}(\theta; x_i, y_i^+, y_i^-)$. This implies $\|\widetilde{g}\|_2 \leq 2\sum_{i=1}^n q_i^\star = 2$. Therefore, we can apply proposition 1 to obtain the following bound.

$$\mathbb{E}\left[\ell_{\text{DPO,TV}}(\overline{\theta}; \mathcal{D}_{\text{src}})\right] - \min_\theta \ell_{\text{DPO,TV}}(\theta; \mathcal{D}_{\text{src}}) \leq \delta + \frac{2}{\sqrt{T}}$$

$$\leq 6\beta(2B + J)(1 + 2\rho) \cdot \sqrt{\frac{4 + \log n}{n}} + \frac{2}{\sqrt{T}}$$

Now if we set $T = O(1/\varepsilon^2)$ and choose minibatch size so that $n/\log n = O\left(\frac{\beta^2(2B+J)^2(1+2\rho)^2}{\varepsilon^2}\right)$, we obtain the upper bound of $O(\varepsilon)$. $\qquad\square$

**Proposition 1** (Restatement of proposition 3 from (Levy et al., 2020), see also (Lan, 2012), Corollary 1)**.** *Let $F : \Theta \to \mathbb{R}$ and $\overline{F} : \Theta \to \mathbb{R}$ satisfy $0 \leq F(\theta) - \overline{F}(\theta) \leq \delta$ for all $\theta \in \Theta$. Assume that $\overline{F}$ is convex and a stochastic gradient estimator $\widetilde{g}$ satisfies $\mathbb{E}\widetilde{g}(\theta) \in \partial\overline{F}(\theta)$ and $\mathbb{E}\|\widetilde{g}(\theta)\|^2 \leq \Gamma^2$ for all $\theta \in \Theta$. Suppose we run stochastic gradient descent with iterates*

$$\theta_{t+1} = \Pi_\Theta(\theta_t - \eta\widetilde{g}(\theta_t)).$$

*and step-size $\eta = \frac{R}{\Gamma\sqrt{T}}$. Then we have,*

$$\mathbb{E}\left[F\left(\frac{1}{T}\sum_{t=1}^T \theta_t\right)\right] - \min_\theta F(\theta) \lesssim \delta + \frac{\Gamma R}{\sqrt{T}}.$$

### A.3. Robust Policy Optimization

Let us define $v(\theta; x, y) = r(x, y) - \beta\log\frac{\pi_\theta(y|x)}{\pi_{\text{ref}}(y|x)}$ and $v(\theta; x) = \mathbb{E}_{y\sim\pi_\theta(\cdot|x)}[v(\theta; x, y)]$. Furthermore, given a divergence function $\phi$ let us define the following objectives.

$$v_\phi(\theta; x^n) = \inf_{q\in\Delta^n : \frac{1}{n}\sum_{i=1}^n \phi(nq_i)\leq\rho} \sum_{i=1}^n q_i \cdot v(\theta; x_i) \tag{8}$$

$$\overline{v}_\phi(\theta; n) = \mathbb{E}_{x^n\sim\mathcal{D}_{\text{src}}^n}[v_\phi(\theta; x^n)]$$

In order to motivate the natural policy gradient step in algorithm 2 let us define the following loss function.

$$L(w, \theta; x^n) = \sum_i q_i^\star \cdot \mathbb{E}_{y\sim\pi_\theta(\cdot|x_i)}\left[\left(v(\theta; x_i, y) - w^\top\nabla_\theta\log\pi_\theta(y_i|x_i)\right)^2\right]$$

where $\{q_i^\star\}_{i=1}^n$ is the optimal solution to the optimization problem define in eq. (8). Let us define the vector $w_{x^n}^{\pi_\theta}$ as the optimal solution of the following optimization problem.

$$w_{x^n}^{\pi_\theta} = \arg\min_w L(w, \theta; x^n) \tag{9}$$

Let us define $G_{x^n}^{\pi_\theta}$ to be the weighted Fisher information matrix i.e.

$$G_{x^n}^{\pi_\theta} = \sum_{i=1}^n q_i^\star \cdot \mathbb{E}_{y\sim\pi_\theta(\cdot|x_i)}\left[\nabla_\theta\log\pi_\theta(y|x_i)\nabla_\theta\log\pi_\theta(y|x_i)^\top\right]$$

Then Lemma 5 shows that $w_{x^n}^{\pi_\theta} = (G_{x^n}^{\pi_\theta})^\dagger \nabla_\theta v_\phi(\theta; x^n)$. In other words, algorithm 2 performs the following gradient steps for policy optimization.

$$\theta_{t+1} = \theta_t + \eta_t \cdot w_t \quad \text{where } w_t \in \arg\min_w L\left(w, \theta; \{x_i\}_{x_i \in B_t}\right)$$

We now follow a Lyapunov potential based approach which is standard in the analysis of natural policy gradient based algorithms (Cayci et al., 2024). Let us define the following potential function.

$$\Phi(\pi) = \sum_x \mathcal{D}_{\text{src}}(x) \sum_y \pi^\star(y|x) \log \frac{\pi^\star(y|x)}{\pi(y|x)}$$

### A.4. Proof of Theorem 2

*Proof.* The main ingredient is Lemma 2, which shows that if the number of iterations $T \geq \log(1/\varepsilon_1 \cdot \log|Y|)$ and the number of minibatch samples $n \geq \frac{\sqrt{\varepsilon_{\text{apx}}}(\beta B^2 + r_{\max})^2}{\eta \beta^2 q_{\min}^2 \sigma^2} \frac{\log(\log\log(|Y|/\varepsilon_1)/\delta)}{\varepsilon_1^2}$ then we have $\Phi(\pi_T) \leq O(\varepsilon_1)$ with probability at least $1 - \delta$. Now lemma 3 implies the following bound on the suboptimality gap.

$$v_{\text{TV}}(\theta^\star; \mathcal{D}_{\text{src}}) - v_{\text{TV}}(\theta_T; \mathcal{D}_{\text{src}}) \leq 2\sqrt{2}(r_{\max} + 3\beta B^2)\sqrt{1 + 4\rho^2/\nu}\sqrt{\varepsilon_1}$$

Now substituting $\varepsilon_1 = \frac{\varepsilon^2}{1 + 4\rho^2/\nu} \frac{1}{8(r_{\max} + 3\beta B^2)^2}$ we obtain the desired bound on the number of iterations $T$, and the minibatch size $n$. $\qquad\square$

The next lemma provides a bound on $\Phi(\pi_T)$ provided the number of iterations $T$ is large (at least $O(\log\log(1/\varepsilon))$), and the number of samples per minibatch $n$ is at least $O(\log\log\log(1/\varepsilon)/\varepsilon^2)$.

**Lemma 2.** *Suppose assumption 3 and assumption 4 hold, and $q_{t,i}^\star \geq q_{\min}$ for all $t$. If the robust policy optimization (algorithm 2) is run for $T \geq O\left(\log\left(\frac{\log|Y|}{\varepsilon}\right)\right)$ iterations, and $n \geq \max\left\{\frac{\sqrt{\varepsilon_{\text{apx}}}}{\varepsilon \cdot q_{\min}^2}, \frac{(\beta B^2 + r_{\max})^2}{\eta \beta^2 q_{\min}^2 \sigma^2} \frac{\log(\log\log(|Y|/\varepsilon)/\delta)}{\varepsilon^2}\right\}$ minibatch samples per iteration. Then the final policy $\pi_T = \pi_{\theta_T}$ satisfies,*

$$\Phi(\pi_T) \leq O(\varepsilon)$$

*with probability at least $1 - \delta$, provided $\eta < \min\left\{\frac{1}{2\beta q_{\min}}, \frac{q_{\min} r_{\min} \sigma^2}{(6\beta B^2 + r_{\max})^2}\right\}$ and $\beta < \frac{r_{\min}}{8B^2}$.*

*Proof.* Lemma 4 proves that the following recurrence holds for each iteration $t$ with probability at least $1 - \delta/t^2$.

$$\Phi(\pi_{t+1}) \leq (1 - \eta\beta q_{\min})\Phi(\pi_t) - \frac{\eta}{n}\Delta_t - \frac{\eta}{n}\sum_i q_{t,i}^\star \cdot v(\theta_t; x_i)$$

$$+ O\left(\sqrt{\frac{(\eta^2\|w_t\|_2^2 + \eta\|w_t\|_2)\log(t/\delta)}{n}}\right) + \frac{\eta^2}{2}\|w_t\|_2^2 + \frac{\eta}{q_{\min} \cdot n}\sqrt{C \cdot L(w_t; \theta_t, \{x_i\}_{i \in B_t})} \quad (10)$$

Therefore, by a union bound over all iterations the above bound holds for all $t \geq 1$. Using assumption 4 and lemma 5 we obtain the following bound on the norm of the vector $w_t$.

$$\|w_t\|_2 \leq \frac{1}{\sigma}\|\nabla_\theta v_\phi(\theta; x^n)\|_2 \leq \frac{1}{\sigma}\sum_{i=1}^n q_{t,i}^\star \|\nabla_\theta v(\theta_t; x_i)\|_2 \leq \frac{1}{\sigma}\sum_{i=1}^n q_{t,i}^\star \left\|\mathbb{E}_{y\sim\pi_{\theta_t}(\cdot|x_i)}[\nabla_\theta \log \pi_{\theta_t}(y|x_i)(v(\theta; x_i, y) - \beta)]\right\|_2$$

Under log-linear policy, $\beta \cdot \mathbb{E}_{y\sim\pi_{\theta_t}(\cdot|x_i)}[\nabla_\theta \log \pi_{\theta_t}(y|x_i)] = \beta \cdot \sum_y \pi_{\theta_t}(y|x_i)\left(\psi(x_i, y) - \sum_{y'}\pi_{\theta_t}(y'|x_i)\psi(x_i, y')\right) = 0$. Therefore, we obtain

$$\|w_t\|_2 \leq \frac{1}{\sigma}\sum_{i=1}^n q_{t,i}^\star \left\|\mathbb{E}_{y\sim\pi_{\theta_t}(\cdot|x_i)}[\nabla_\theta \log \pi_{\theta_t}(y|x_i)v(\theta_t; x_i, y)]\right\|_2 \quad (11)$$

Now recall, that $v(\theta_t; x_i, y) = r(x_i, y) - \beta \log \frac{\pi_{\theta_t}(y|x_i)}{\pi_{\text{ref}}(y|x_i)}$. If $\pi_{\text{ref}}$ is also log-linear then from the smoothness of $\log \pi_\theta(y|x)$ (e.g. (Agarwal et al., 2021), lemma 34) we obtain the following bound.

$$\log \frac{\pi_{\theta_t}(y|x_i)}{\pi_{\text{ref}}(y|x_i)} \leq \|\theta_t - \theta_{\text{ref}}\|_2^2 - \nabla_\theta \log \pi_{\theta_t}(y|x_i)^\top (\theta_{\text{ref}} - \theta_t) \leq 2B^2 + 4B$$

The last inequality uses $\|\nabla_\theta \log \pi_\theta(y|x_i)\|_2 \leq 2$. Therefore, for any $(x_i, y)$ and $\theta$ we have $|v(\theta; x_i, y)| \leq 6\beta B^2 + r_{\max}$. Substituting this bound in eq. (11), we get the following bound on the norm of $w_t$.

$$\|w_t\|_2 \leq \frac{6\beta B^2 + r_{\max}}{\sigma} \tag{12}$$

We now establish a lower bound on the term $\sum_i q_{t,i}^\star \cdot v(\theta_t; x_i)$.

$$v(\theta_t; x_i) = \sum_y \pi_{\theta_t}(y|x_i) \left( r(x_i, y) - \beta \log \frac{\pi_{\theta_t}(y|x_i)}{\pi_{\text{ref}}(y|x_i)} \right) \geq r_{\min} - 4\beta B^2 \geq \frac{r_{\min}}{2}$$

as long as $\beta \leq \frac{r_{\min}}{8B^2}$. Therefore, we obtain the following lower bound.

$$\sum_i q_{t,i}^\star \cdot v(\theta_t; x_i) \geq \frac{r_{\min}}{2} \cdot nq_{\min} \tag{13}$$

Substituting the bounds of eq. (12) and eq. (13) in eq. (10) and rearranging we obtain the following recurrence relation.

$$\Phi(\pi_{t+1}) \leq (1 - \eta\beta q_{\min}) \Phi(\pi_t) - \frac{\eta}{n}\Delta_t - \eta q_{\min}\frac{r_{\min}}{2}$$
$$+ O\left( \frac{\beta B^2 + r_{\max}}{\sigma} \sqrt{\frac{\eta \log(t/\delta)}{n}} \right) + \frac{\eta^2}{2}\left( \frac{6\beta B^2 + r_{\max}}{\sigma} \right)^2 + \frac{\eta}{q_{\min} \cdot n} \sqrt{C \cdot L(w_t; \theta_t, \{x_i\}_{i \in B_t})} \tag{14}$$

If $\eta \leq \frac{q_{\min} r_{\min} \sigma^2}{(6\beta B^2 + r_{\max})^2}$ then we have,

$$-\eta q_{\min}\frac{r_{\min}}{2} + \frac{\eta^2}{2}\left( \frac{6\beta B^2 + r_{\max}}{\sigma} \right)^2 \leq 0$$

and we obtain a new recurrence relation.

$$\Phi(\pi_{t+1}) \leq (1 - \eta\beta q_{\min}) \Phi(\pi_t) - \frac{\eta}{n}\Delta_t + \frac{\eta}{n \cdot q_{\min}} \sqrt{\epsilon_{\text{apx}}} + O\left( \frac{\beta B^2 + r_{\max}}{\sigma} \sqrt{\frac{\eta \log(t/\delta)}{n}} \right) \tag{15}$$

Now using $\Phi(\pi_0) \leq \log |Y|$ and induction we obtain the following upper bound.

$$\Phi(\pi_{t+1}) \leq (1 - \eta\beta q_{\min})^{t+1} \log |Y| - \frac{\eta}{n} \sum_{k=0}^t (1 - \eta\beta q_{\min})^{t-k} \Delta_k$$

$$+ \eta \sum_{k=0}^t (1 - \eta\beta q_{\min})^{t-k} \frac{1}{n \cdot q_{\min}} \sqrt{\epsilon_{\text{apx}}}$$

$$+ \sqrt{\eta} \sum_{k=0}^t (1 - \eta\beta q_{\min})^{t-k} O\left( \frac{\beta B^2 + r_{\max}}{\sigma} \sqrt{\frac{\log(k/\delta)}{n}} \right)$$

Therefore, for any $t \leq T$ we obtain the following upper bound on the potential function.

$$\Phi(\pi_{t+1}) \leq (1 - \eta\beta q_{\min})^{t+1} \log |Y| + \frac{\sqrt{\epsilon_{\text{apx}}}}{n \cdot q_{\min}^2} + O\left( \frac{1}{\sqrt{\eta}\beta q_{\min}} \frac{\beta B^2 + r_{\max}}{\sigma} \sqrt{\frac{\log(T/\delta)}{n}} \right)$$

In particular, if we choose $\eta$ such that $\eta < \frac{1}{2\beta q_{\min}}$ and $T \geq O\left(\log\left(\frac{\log|Y|}{\varepsilon}\right)\right)$ then we get

$$\Phi(\pi_T) \leq O(\varepsilon) + \frac{\sqrt{\epsilon_{\mathrm{apx}}}}{n \cdot q_{\min}^2} + O\left(\frac{1}{\sqrt{\eta}\beta q_{\min}}\frac{\beta B^2 + r_{\max}}{\sigma}\sqrt{\frac{\log(\log\log(|Y|/\varepsilon)/\delta)}{n}}\right)$$

Finally, choosing $n \geq \max\left\{\frac{\sqrt{\epsilon_{\mathrm{apx}}}}{\varepsilon \cdot q_{\min}^2}, \frac{(\beta B^2 + r_{\max})^2}{\eta\beta^2 q_{\min}^2 \sigma^2}\frac{\log(\log\log(|Y|/\varepsilon)/\delta)}{\varepsilon^2}\right\}$ gives us $\Phi(\pi_T) \leq \varepsilon$.

$\square$

The next lemma shows that if the potential $\Phi(\pi)$ of a policy is small, then the policy $\pi$ is approximately optimal.

**Lemma 3.** *Suppose $\Phi(\pi_\theta) \leq \varepsilon$ and $\mathcal{D}_{\mathrm{src}}(x) \geq \nu$ for any $x$. Then*

$$v_{\mathrm{TV}}(\theta^\star; \mathcal{D}_{\mathrm{src}}) - v_{\mathrm{TV}}(\theta; \mathcal{D}_{\mathrm{src}}) \leq 2\sqrt{2}\left(r_{\max} + 3\beta B^2\right)\sqrt{(1 + 4\rho^2/\nu)\varepsilon}.$$

*Proof.*

$$v_{\mathrm{TV}}(\theta^\star; \mathcal{D}_{\mathrm{src}}) - v_{\mathrm{TV}}(\theta; \mathcal{D}_{\mathrm{src}}) = \inf_{\mathcal{D}:d_{\mathrm{TV}}(\mathcal{D},\mathcal{D}_{\mathrm{src}})\leq\rho} v(\theta^\star; \mathcal{D}) - \inf_{\mathcal{D}:d_{\mathrm{TV}}(\mathcal{D},\mathcal{D}_{\mathrm{src}})\leq\rho} v(\theta; \mathcal{D})$$

Now suppose $\mathcal{D}_0$ attains the optimal value in the second optimization problem. Then we have,

$$
\begin{aligned}
&v_{\mathrm{TV}}(\theta^\star; \mathcal{D}_{\mathrm{src}}) - v_{\mathrm{TV}}(\theta; \mathcal{D}_{\mathrm{src}}) \\
&\leq v(\theta^\star; \mathcal{D}_0) - v(\theta; \mathcal{D}_0) \\
&= \sum_x \mathcal{D}_0(x)\left(\mathbb{E}_{y\sim\pi_{\theta^\star}(\cdot|x)}[v(\theta^\star; x, y)] - \mathbb{E}_{y\sim\pi_\theta(\cdot|x)}[v(\theta; x, y)]\right) \\
&= \sum_x \mathcal{D}_0(x)\left(\sum_y (\pi_{\theta^\star}(y|x) - \pi_\theta(y|x))r(x, y)\right) \\
&\quad - \beta\sum_x \mathcal{D}_0(x)\sum_y\left(\pi_{\theta^\star}(y|x)\log\frac{\pi_{\theta^\star}(y|x)}{\pi_{\mathrm{ref}}(y|x)} - \pi_\theta(y|x)\log\frac{\pi_\theta(y|x)}{\pi_{\mathrm{ref}}(y|x)}\right) \\
&\leq 2r_{\max}\sum_x \mathcal{D}_0(x)\cdot d_{\mathrm{TV}}\left(\pi_{\theta^\star}(\cdot|x), \pi_\theta(\cdot|x)\right) - \beta\sum_x \mathcal{D}_0(x)d_{\mathrm{KL}}\left(\pi_{\theta^\star}(\cdot|x), \pi_\theta(\cdot|x)\right) \\
&\quad - \beta\sum_x \mathcal{D}_0(x)\sum_y (\pi_{\theta^\star}(y|x) - \pi_\theta(y|x))\log\frac{\pi_\theta(y|x)}{\pi_{\mathrm{ref}}(y|x)}
\end{aligned}
$$

If $\pi_{\mathrm{ref}}$ is also log-linear then from the smoothness of $\log\pi_\theta(y|x)$ (e.g. (Agarwal et al., 2021), lemma 34) we obtain the following bound.

$$\log\frac{\pi_\theta(y|x_i)}{\pi_{\mathrm{ref}}(y|x_i)} \leq \|\theta - \theta_{\mathrm{ref}}\|_2^2 - \nabla_\theta\log\pi_\theta(y|x_i)^\top(\theta_{\mathrm{ref}} - \theta) \leq 2B^2 + 4B$$

This gives us the following upper bound.

$$
v_{\text{TV}}(\theta^\star; \mathcal{D}_{\text{src}}) - v_{\text{TV}}(\theta; \mathcal{D}_{\text{src}})
$$

$$
\leq 2r_{\max} \sum_x \mathcal{D}_0(x) \cdot d_{\text{TV}} \left( \pi_{\theta^\star}(\cdot|x), \pi_\theta(\cdot|x) \right) - \beta \sum_x \mathcal{D}_0(x) d_{\text{KL}} \left( \pi_{\theta^\star}(\cdot|x), \pi_\theta(\cdot|x) \right)
$$

$$
+ 6\beta B^2 \sum_x \mathcal{D}_0(x) \cdot d_{\text{TV}} \left( \pi_{\theta^\star}(\cdot|x), \pi_\theta(\cdot|x) \right)
$$

$$
\leq \left( 2r_{\max} + 6\beta B^2 \right) \sum_x \mathcal{D}_0(x) \cdot d_{\text{TV}} \left( \pi_{\theta^\star}(\cdot|x), \pi_\theta(\cdot|x) \right)
$$

$$
\leq \left( 2r_{\max} + 6\beta B^2 \right) \sqrt{\sum_x \frac{\mathcal{D}_0^2(x)}{\mathcal{D}_{\text{src}}(x)}} \sqrt{\sum_x \mathcal{D}_{\text{src}}(x) \cdot d_{\text{TV}}^2 \left( \pi_{\theta^\star}(\cdot|x), \pi_\theta(\cdot|x) \right)}
$$

$$
\leq 2\sqrt{2} \left( r_{\max} + 3\beta B^2 \right) \sqrt{1 + d_{\chi^2}(\mathcal{D}_0, \mathcal{D}_{\text{src}})} \sqrt{\sum_x \mathcal{D}_{\text{src}}(x) \cdot d_{\text{KL}} \left( \pi_{\theta^\star}(\cdot|x), \pi_\theta(\cdot|x) \right)}
$$

$$
= 2\sqrt{2} \left( r_{\max} + 3\beta B^2 \right) \sqrt{1 + d_{\chi^2}(\mathcal{D}_0, \mathcal{D}_{\text{src}})} \sqrt{\Phi(\pi_\theta)}
$$

The last inequality uses Pinsker's inequality. Now the bound follows from the following observation. $d_{\chi^2}(\mathcal{D}_0, \mathcal{D}_{\text{src}}) \leq \frac{1}{\nu} \sum_x (\mathcal{D}_0(x) - \mathcal{D}_{\text{src}}(x))^2 \leq \frac{1}{\nu} (d_{\text{TV}}(\mathcal{D}_0, \mathcal{D}_{\text{src}}))^2 \leq 4\rho^2/\nu.$ $\qquad\square$

**Lemma 4.** *Suppose $q_{t,i}^\star \geq q_{\min}$, and assumption 3 holds. Then the sequence of policies $\{\pi_t = \pi_{\theta_t}\}_{t \geq 1}$ generated by algorithm 2 satisfy the following recurrence relation with probability at least $1 - \delta/t^2$,*

$$
\Phi(\pi_{t+1}) \leq (1 - \eta\beta q_{\min}) \Phi(\pi_t) - \frac{\eta}{n} \Delta_t - \frac{\eta}{n} \sum_i q_{t,i}^\star \cdot v(\theta_t; x_i)
$$

$$
+ O\left( \sqrt{\frac{(\eta^2 \|w_t\|_2^2 + \eta \|w_t\|_2) \log(t/\delta)}{n}} \right) + \frac{\eta^2}{2} \|w_t\|_2^2 + \frac{\eta}{q_{\min} \cdot n} \sqrt{C \cdot L(w_t; \theta_t, \{x_i\}_{i \in B_t})}
$$

*where $\Delta_t = v_\phi(\theta^\star; x^n) - v_\phi(\theta_t; x^n)$.*

*Proof.* First, using log-linear policy class (assumption 2) and smoothness of $\log \pi_\theta(y|x)$ (c.f. (Agarwal et al., 2021), lemma 34) we obtain the following bound.

$$
\log \frac{\pi_t(y|x)}{\pi_{t+1}(y|x)} = \log \frac{\pi_{\theta_t}(y|x)}{\pi_{\theta_t + \eta w_t}(y|x)} \leq \frac{\eta^2}{2} \|w_t\|_2^2 - \eta \nabla_\theta \log \pi_t(y|x)^\top w_t \tag{16}
$$

$$
\Phi(\pi_{t+1}) - \Phi(\pi_t) = \sum_x \mathcal{D}_{\text{src}}(x) \sum_y \pi^\star(y|x) \log \frac{\pi_t(y|x)}{\pi_{t+1}(y|x)}
$$

$$
\leq O\left( \sqrt{\frac{(\eta^2 \|w_t\|_2^2 + \eta \|w_t\|_2) \log(t/\delta)}{n}} \right) + \frac{1}{n} \sum_{i=1}^n \sum_y \pi^\star(y|x_i) \log \frac{\pi_t(y|x_i)}{\pi_{t+1}(y|x_i)}
$$

$$
\leq O\left( \sqrt{\frac{(\eta^2 \|w_t\|_2^2 + \eta \|w_t\|_2) \log(t/\delta)}{n}} \right) + \eta^2 \frac{\|w_t\|_2^2}{2} - \frac{\eta}{n} \sum_i \sum_y \pi^\star(y|x_i) \nabla_\theta^\top \log \pi_t(y|x) w_t
$$

$$
= O\left( \sqrt{\frac{(\eta^2 \|w_t\|_2^2 + \eta \|w_t\|_2) \log(t/\delta)}{n}} \right) + \eta^2 \frac{\|w_t\|_2^2}{2} \underbrace{- \frac{\eta}{n} \sum_i \sum_y \pi^\star(y|x_i) \left( \nabla_\theta^\top \log \pi_t(y|x) w_t - v(\theta; x_i, y) \right)}_{:=T_1}
$$

$$
- \frac{\eta}{n} \sum_i \sum_y \pi^\star(y|x_i) v(\theta; x_i, y) - \frac{\eta}{n} \Delta_t + \frac{\eta}{n} \Delta_t \tag{17}
$$

The first inequality uses two facts (1) The set $B_t = \{x_i : i \in [n]\}$ is a uniformly random drawn sample of size $n$ from the distribution $\mathcal{D}_{\mathrm{src}}$ and for each $x$, and (2) for each $x$, the term $\sum_y \pi^\star(y|x) \log \frac{\pi_t(y|x)}{\pi_{t+1}(y|x)} \leq O(\eta^2 \|w_t\|_2^2 + \eta \|w_t\|_2)$. Therefore, we can use Chernoff-Hoeffding bound and get a bound on the approximation error i.e.

$$\mathbb{P}\left(\sum_x \mathcal{D}_{\mathrm{src}}(x) \sum_y \pi^\star(y|x) \log \frac{\pi_t(y|x)}{\pi_{t+1}(y|x)} - \frac{1}{n} \sum_{i=1}^n \sum_y \pi^\star(y|x_i) \log \frac{\pi_t(y|x_i)}{\pi_{t+1}(y|x_i)} \geq O\left(\sqrt{\frac{(\eta^2 \|w_t\|_2^2 + \eta \|w_t\|_2) \log(t/\delta)}{n}}\right)\right) \leq \frac{\delta}{t^2}$$

The second inequality uses eq. (16), and the final equality in the derivation of eq. (17) substitutes $\Delta_t = v_\phi(\theta^\star; x^n) - v_\phi(\theta_t; x^n)$. We bound the term $T_1$ using assumption 3.

$$
\begin{aligned}
T_1 &\leq \frac{\eta}{n} \sum_i \sum_y \pi^\star(y|x_i) \cdot \left|\nabla_\theta^\top \log \pi_t(y|x) w_t - v(\theta; x_i, y)\right| \\
&\leq \frac{\eta}{q_{\min} \cdot n} \sum_i q_{t,i}^\star \sum_y \pi^\star(y|x_i) \cdot \left|\nabla_\theta^\top \log \pi_t(y|x) w_t - v(\theta; x_i, y)\right| \\
&\leq \frac{\eta}{q_{\min} \cdot n} \sum_i q_{t,i}^\star \sum_y \frac{\pi^\star(y|x_i)}{\sqrt{\pi_t(y|x_i)}} \cdot \sqrt{\pi_t(y|x_i)} \cdot \left|\nabla_\theta^\top \log \pi_t(y|x) w_t - v(\theta; x_i, y)\right| \\
&\leq \frac{\eta}{q_{\min} \cdot n} \sum_i q_{t,i}^\star \sqrt{\sum_y \pi_t(y|x_i) \left(\frac{\pi^\star(y|x_i)}{\pi_t(y|x_i)}\right)^2} \sqrt{\sum_y \pi_t(y|x_i) \left(\nabla_\theta^\top \log \pi_t(y|x) w_t - v(\theta; x_i, y)\right)^2} \\
&\leq \frac{\eta}{q_{\min} \cdot n} \sqrt{\sum_i q_{t,i}^\star \cdot \mathbb{E}_{y \sim \pi_t(\cdot|x_i)}\left[\left(\frac{\pi^\star(y|x_i)}{\pi_t(y|x_i)}\right)^2\right]} \sqrt{\sum_i q_{t,i}^\star \cdot \mathbb{E}_{y \sim \pi_t(y|x_i)}\left[\left(\nabla_\theta^\top \log \pi_t(y|x) w_t - v(\theta; x_i, y)\right)^2\right]} \\
&\leq \frac{\eta}{q_{\min} \cdot n} \sqrt{C \cdot L(w_t; \theta_t, \{x_i\}_{i \in B_t})}
\end{aligned}
$$

We now bound the term $\Delta_t$.

$$
\begin{aligned}
\Delta_t &= (v_\phi(\theta^\star, x^n) - v_\phi(\theta_t; x^n)) \\
&= \inf_{q \in \Delta^n; d_{\mathrm{TV}}(q, 1/n) \leq \rho} \sum_i q_i \cdot v(\theta^\star; x_i) - \inf_{q \in \Delta^n; d_{\mathrm{TV}}(q, 1/n) \leq \rho} \sum_i q_i \cdot v(\theta_t; x_i) \\
&\leq \sum_i q_{t,i}^\star (v(\theta^\star, x_i) - v(\theta_t, x_i)) \\
&= \sum_i q_{t,i}^\star \sum_y \pi_{\theta^\star}(y|x_i) \left(r(x_i, y) - \beta \log \frac{\pi_{\theta^\star}(y|x_i)}{\pi_{\mathrm{ref}}(y|x_i)} - v(\theta_t; x_i)\right)
\end{aligned}
$$

The first inequality follows from the fact that $\{q_{t,i}^\star\}_{i \in [n]}$ optimizes the second minimization problem. Now let us define the following notion of a soft advantage function.

$$A(\theta_t; x_i, y) = r(x_i, y) - \beta \log \frac{\pi_{\theta_t}(y|x_i)}{\pi_{\mathrm{ref}}(y|x_i)} - v(\theta_t; x_i) \tag{18}$$

Then we obtain the following upper bound on $\Delta_t$.

$$
\begin{aligned}
\Delta_t &\leq \sum_i q_{t,i}^\star \sum_y \pi_{\theta^\star}(y|x_i) \left(A(\theta_t; x_i, y) - \beta \log \frac{\pi_{\theta^\star}(y|x_i)}{\pi_{\theta_t}(y|x_i)}\right) \\
&\leq \sum_i q_{t,i}^\star \sum_y \pi_{\theta^\star}(y|x_i) A(\theta_t; x_i, y) - n\beta q_{\min} \frac{1}{n} \sum_{i=1}^n \sum_y \pi_{\theta^\star}(y|x_i) \log \frac{\pi_{\theta^\star}(y|x_i)}{\pi_t(y|x_i)} \\
&\leq \sum_i q_{t,i}^\star \sum_y \pi_{\theta^\star}(y|x_i) A(\theta_t; x_i, y) - n\beta q_{\min} \left(\Phi(\pi_t) - O\left(\sqrt{\frac{\log(t/\delta)}{n}}\right)\right)
\end{aligned}
$$

The last inequality uses the fact that the set $B_t = \{x_i : i \in [n]\}$ is a uniformly random set of size $n$ and one can apply Chernoff-Hoeffding bound to replace the sample average with the mean $\Phi(\pi_t)$ and an additive error of $\widetilde{O}(1/\sqrt{n})$.

Substituting the upper bound above in eq. (17) we obtain the following upper bound on the potential difference.

$$
\Phi(\pi_{t+1}) - \Phi(\pi_t) \leq O\left(\sqrt{\frac{(\eta^2 \|w_t\|_2^2 + \eta \|w_t\|_2) \log(t/\delta)}{n}}\right) + \frac{\eta^2}{2}\|w_t\|_2^2 + \frac{\eta}{q_{\min} \cdot n}\sqrt{C \cdot L(w_t; \theta_t, \{x_i\}_{i \in B_t})}
$$
$$
- \frac{\eta}{n}\sum_i \sum_y \pi^\star(y|x_i) v(\theta; x_i, y) - \frac{\eta}{n}\Delta_t + \frac{\eta}{n}\sum_i q_{t,i}^\star \sum_y \pi_{\theta^\star}(y|x_i) A(\theta_t; x_i, y) - \eta\beta q_{\min} \cdot \Phi(\pi_t)
$$
$$
\leq O\left(\sqrt{\frac{(\eta^2 \|w_t\|_2^2 + \eta \|w_t\|_2) \log(t/\delta)}{n}}\right) + \frac{\eta^2}{2}\|w_t\|_2^2 + \frac{\eta}{q_{\min} \cdot n}\sqrt{C \cdot L(w_t; \theta_t, \{x_i\}_{i \in B_t})}
$$
$$
- \frac{\eta}{n}\sum_i q_{t,i}^\star \sum_y \pi^\star(y|x_i) v(\theta_t; x_i, y) - \frac{\eta}{n}\Delta_t + \frac{\eta}{n}\sum_i q_{t,i}^\star \sum_y \pi_{\theta^\star}(y|x_i) A(\theta_t; x_i, y) - \eta\beta q_{\min} \cdot \Phi(\pi_t)
$$
$$
\leq O\left(\sqrt{\frac{(\eta^2 \|w_t\|_2^2 + \eta \|w_t\|_2) \log(t/\delta)}{n}}\right) + \frac{\eta^2}{2}\|w_t\|_2^2 + \frac{\eta}{q_{\min} \cdot n}\sqrt{C \cdot L(w_t; \theta_t, \{x_i\}_{i \in B_t})}
$$
$$
- \frac{\eta}{n}\sum_i q_{t,i}^\star \cdot v(\theta_t; x_i) - \frac{\eta}{n}\Delta_t - \eta\beta q_{\min} \cdot \Phi(\pi_t)
$$

The last inequality uses the definition of advantage function eq. (18).

$\square$

**Lemma 5.** *Let $\{q_i^\star\}_{i=1}^n$ be the optimal solution to the optimization problem eq. (8).*

$$
w_{x^n}^{\pi_\theta} = \left(\sum_{i=1}^n q_i^\star \cdot \mathop{\mathbb{E}}_{y \sim \pi_\theta(\cdot|x_i)}\left[\nabla_\theta \log \pi_\theta(y|x_i) \nabla_\theta \log \pi_\theta(y|x_i)^\top\right]\right)^\dagger \nabla_\theta v_\phi(\theta; x^n)
$$

*Proof.* From the first-order optimality conditions related to the optimization problem defined in eq. (9), we have

$$
\sum_i q_i^\star \cdot \mathop{\mathbb{E}}_{y \sim \pi_\theta(\cdot|x_i)}\left[\nabla_\theta \log \pi_\theta(y_i|x)\left(v(\theta; x_i, y) - w^\top \nabla_\theta \log \pi_\theta(y_i|x)\right)\right] = 0 \tag{19}
$$

Moreover, since $\{q_i^\star\}_{i=1}^n$ are the optimal weights we have

$$
\nabla_\theta v_\phi(\theta; x^n) = \sum_{i=1}^n q_i^\star \cdot \nabla_\theta v(\theta; x_i) = \sum_{i=1}^n q_i^\star \cdot \mathop{\mathbb{E}}_{y \sim \pi_\theta(\cdot|x_i)}\left[\nabla_\theta \log \pi_\theta(y|x_i)\left(v(\theta; x_i, y) - \beta\right)\right]
$$
$$
= \sum_{i=1}^n q_i^\star \cdot \mathop{\mathbb{E}}_{y \sim \pi_\theta(\cdot|x_i)}\left[\nabla_\theta \log \pi_\theta(y|x_i) v(\theta; x_i, y)\right]
$$

The last equality uses the fact that under log-linear policy,

$$
\beta \mathbb{E}_{y \sim \pi_\theta(\cdot|x_i)}[\nabla_\theta \log \pi_\theta(y|x_i)] = \beta \cdot \sum_y \pi_\theta(y|x_i)\left(\psi(x_i, y) - \sum_{y'} \pi_\theta(y'|x_i)\psi(x_i, y')\right) = 0.
$$

Now rearranging eq. (19) we get the desired expression for the vector $w_{x^n}^{\pi_\theta}$. $\square$

# B. Experiment Details

We train our models on a single machine with $8\times$H100 GPUs. Approx. time to train `google/gemma-2b-it`:

- a reward model (with the 400k dataset): 6 h
- PPO policy (with the 5k dataset): 10 min

- DPO policy (with the 400k dataset): 5 h

For training we use the `trl` library by Hugging Face.

The tables show experiment results with the `google/gemma-2b-it` model, unless stated otherwise.

Note: in the provided code $\rho$ is called `EPS` or `eps`, and DR PPO *version B* is called `solve_pilossgrad` or *solve π loss gradient*, due to an earlier version of the manuscript.

## B.1. Reward Models

We do not perform any hyperparameter tuning for the reward models, and use values suggested by Yang et al. (2024b). We train the reward models with the hyperparameters (otherwise using defaults from `trl`) indicated in Table 6.

*Table 6.* Hyperparameters used for reward model experiments.

| Hyperparameter | Value |
|---|---|
| Base model | `google/gemma-2b-it` |
| Number of epochs | 2 |
| Learning rate $\eta$ | 1e-5 |
| Torch dtype | `bfloat16` |
| Attention implementation | `flash_attention_2` |
| Per device train batch size | 16 |
| Learning rate scheduler type | cosine |
| Warmup ratio | 0.03 |
| Weight decay | 0 |
| Optimizer | `adamw_hf` |
| Maximum sequence length | 1024 |
| LoRA task type | `SEQ_CLS` |
| LoRA $r$ | 32 |
| LoRA alpha | 64 |
| LoRA dropout | 0.05 |

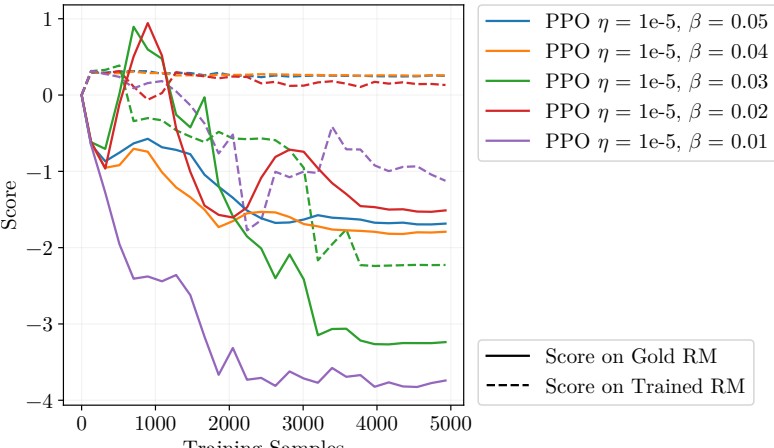

*Figure 1.* Choice of KL regularization coefficient $\beta$ for PPO experiments.

*Table 7.* Evaluation of the reward models on Unified-Feedback (ID, Jiang et al., 2023), HHH Alignment (OOD, Askell et al., 2021), and MT Bench (OOD, Zheng et al., 2023) benchmarks. Trained on a 400K item subset of the Unified-Feedback dataset (Jiang et al., 2023) for 2 epochs. Comparison of different dataset versions – unfiltered indicates the training dataset was used as is; no hh-rlhf indicates the hh-rlhf subset was filtered out; no chatbot_arena_conversations indicates the chatbot_arena_conversations subset was filtered out.

| Reward Model | Unified-Feedback | HHH Alignment | MT Bench |
|---|---|---|---|
| unfiltered; TV dist. | | | |
| BTL RM | 76.5 | 83.3 | 72.7 |
| DR RM $\rho = 0.1$ | 76.9 | 83.3 | 73.6 |
| DR RM $\rho = 0.2$ | 76.4 | 82.4 | 72.9 |
| DR RM $\rho = 0.4$ | 76.2 | 81.0 | 72.9 |
| DR RM $\rho = 0.6$ | 76.0 | 83.8 | 73.5 |
| DR RM $\rho = 0.8$ | 75.6 | 82.4 | 72.1 |
| DR RM $\rho = 1.0$ | 75.4 | 83.3 | 71.9 |
| no hh-rlhf; TV dist. | | | |
| BTL RM | 74.3 | 78.2 | 75.8 |
| DR RM $\rho = 0.02$ | 75.3 | 77.8 | 73.5 |
| DR RM $\rho = 0.1$ | 75.7 | 79.6 | 73.2 |
| DR RM $\rho = 0.2$ | 75.6 | 80.6 | 74.4 |
| DR RM $\rho = 0.4$ | 75.8 | 76.9 | 74.1 |
| DR RM $\rho = 0.6$ | 75.3 | 78.2 | 72.7 |
| DR RM $\rho = 0.8$ | 75.6 | 80.6 | 74.9 |
| DR RM $\rho = 1.0$ | 72.5 | 74.1 | 69.0 |
| no hh-rlhf; $\chi^2$ dist. | | | |
| BTL RM | 75.5 | 79.6 | 73.6 |
| DR RM $\rho = 0.05$ | 75.4 | 80.1 | 72.7 |
| DR RM $\rho = 0.1$ | 75.4 | 79.2 | 72.2 |
| DR RM $\rho = 0.2$ | 75.8 | 78.7 | 74.4 |
| DR RM $\rho = 0.3$ | 75.3 | 78.2 | 72.7 |
| DR RM $\rho = 0.4$ | 73.6 | 76.4 | 71.8 |
| DR RM $\rho = 0.5$ | 72.1 | 75.5 | 70.2 |
| no chatbot_arena_conversations; TV dist. | | | |
| BTL RM | 76.1 | 83.8 | 73.1 |
| DR RM $\rho = 0.02$ | 76.2 | 82.9 | 72.9 |
| DR RM $\rho = 0.1$ | 75.6 | 84.3 | 72.9 |
| DR RM $\rho = 0.2$ | 75.9 | 84.3 | 72.4 |
| DR RM $\rho = 0.4$ | 74.0 | 80.6 | 72.1 |
| DR RM $\rho = 0.6$ | 71.0 | 76.4 | 69.4 |
| DR RM $\rho = 0.8$ | 72.0 | 75.9 | 69.8 |
| DR RM $\rho = 1.0$ | 71.4 | 75.9 | 69.8 |

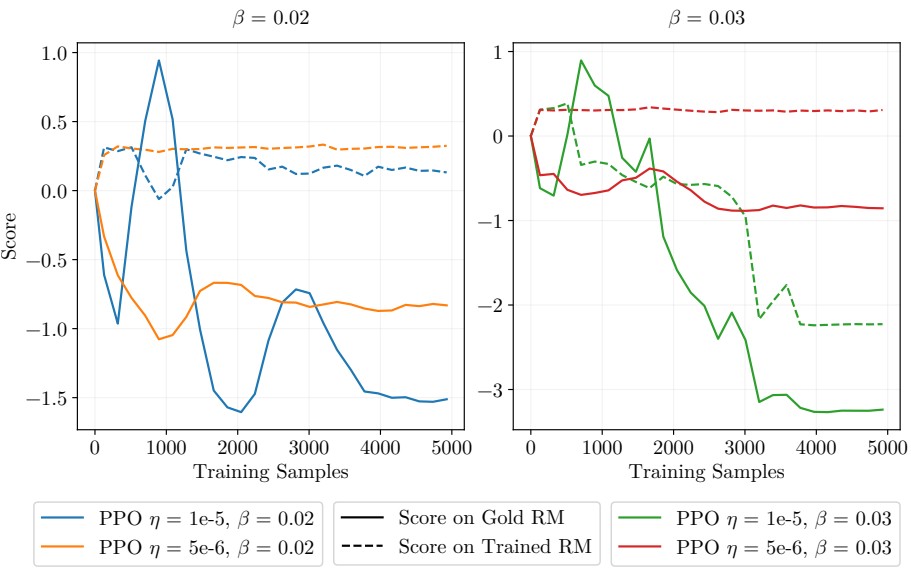

*Figure 2.* Choice of learning rate $\eta$ for PPO experiments.

*Table 8.* Evaluation of the reward models on the RewardBench (Lambert et al., 2024) benchmark. Trained on a 400K item subset of the Unified-Feedback dataset (Jiang et al., 2023) for 2 epochs, with the TV distance. Base model used for training: `mistralai/Mistral-7B-Instruct-v0.2`.

| Reward Model | Chat | Chat-Hard | Safety | Reasoning | Avg. |
|---|---|---|---|---|---|
| Base model: `mistralai/Mistral-7B-Instruct-v0.2` | | | | | |
| BTL RM | 97.5 | 61.4 | 86.8 | 71.5 | 79.3 |
| DR RM $\rho = 0.1$ | 97.5 | 59.4 | 87.4 | 66.0 | 77.6 |
| DR RM $\rho = 0.2$ | 97.5 | 56.1 | 86.9 | 74.6 | 78.8 |
| DR RM $\rho = 0.4$ | 90.8 | 46.3 | 83.4 | 56.0 | 69.1 |
| DR RM $\rho = 0.6$ | 88.3 | 40.1 | 80.7 | 58.0 | 66.8 |
| DR RM $\rho = 0.8$ | 84.1 | 42.5 | 76.9 | 49.3 | 63.2 |
| DR RM $\rho = 1.0$ | 83.0 | 46.1 | 76.6 | 59.9 | 66.4 |

### B.1.1. SUBSETS ON REWARDBENCH

The subset names are:

**Chat**
1. `alpacaeval-easy`
2. `alpacaeval-length`
3. `alpacaeval-hard`
4. `mt-bench-easy`
5. `mt-bench-med`

**Chat-Hard**
6. `mt-bench-hard`
7. `llmbar-natural`
8. `llmbar-adver-neighbor`
9. `llmbar-adver-GPTInst`
10. `llmbar-adver-GPTOut`
11. `llmbar-adver-manual`

**Safety**
12. `refusals-dangerous`
13. `refusals-offensive`
14. `xstest-should-refuse`
15. `xstest-should-respond`
16. `donotanswer`

**Reasoning**
17. `math-prm`
18. `hep-cpp`
19. `hep-go`
20. `hep-java`
21. `hep-js`
22. `hep-python`
23. `hep-rust`

### B.2. PPO

We perform minimal hyperparameter search for selecting the best learning rate and KL coefficient. The effect of KL coefficient $\beta$ can be seen in Figure 1. We chose $\beta = 0.02$ and $\beta = 0.03$ as learning was displaying some improvement on the gold reward model. Further, we tune the learning rate $\eta$, this can be seen in Figure 2. We selected $\eta = 1e - 5$ which displayed some learning in the initial training steps.

For other hyperparameters we use the values similar to the ones used for the reward models, see Table 14.

*Table 9.* Evaluation of the distributionally robust reward models on the Reasoning task from the RewardBench (Lambert et al., 2024) benchmark. Trained on a 400K item subset of the Unified-Feedback dataset (Jiang et al., 2023) for 2 epochs, with the TV distance. Base model used for training: `mistralai/Mistral-7B-Instruct-v0.2`. Subsets are detailed in Table 10.

| Reward Model | Reasoning Subsets | | | | | | |
|---|---|---|---|---|---|---|---|
| Base model: `mistralai/Mistral-7B-Instruct-v0.2` | | | | | | | |
| BTL RM | 51.0 | 92.7 | 93.3 | 92.1 | 92.7 | 92.1 | 89.0 |
| DR RM $\rho = 0.1$ | 39.8 | 93.3 | 95.1 | 93.9 | 92.7 | 87.8 | 90.2 |
| DR RM $\rho = 0.2$ | 57.9 | 88.4 | 93.9 | 93.3 | 90.9 | 87.8 | 92.7 |
| DR RM $\rho = 0.4$ | 52.3 | 61.0 | 54.9 | 62.8 | 62.2 | 62.2 | 54.3 |
| DR RM $\rho = 0.6$ | 63.8 | 51.8 | 53.0 | 48.2 | 57.9 | 54.3 | 48.2 |
| DR RM $\rho = 0.8$ | 43.4 | 54.3 | 56.1 | 55.5 | 50.0 | 57.3 | 57.9 |
| DR RM $\rho = 1.0$ | 64.7 | 59.8 | 54.9 | 52.4 | 53.7 | 51.8 | 57.9 |

*Table 10.* Evaluation of the reward models on RewardBench (Lambert et al., 2024). Trained on a **unfiltered** 400K item subset of the Unified-Feedback dataset (Jiang et al., 2023) for 2 epochs using **TV dist.**

| Reward Model | Chat | | | | | Chat-Hard | | | | | | Safety | | | | | Reasoning | | | | | | | Avg. |
|---|---|---|---|---|---|---|---|---|---|---|---|---|---|---|---|---|---|---|---|---|---|---|---|---|
| | 1 | 2 | 3 | 4 | 5 | 6 | 7 | 8 | 9 | 10 | 11 | 12 | 13 | 14 | 15 | 16 | 17 | 18 | 19 | 20 | 21 | 22 | 23 | |
| Base model: `google/gemma-2b-it` | | | | | | | | | | | | | | | | | | | | | | | | |
| BTL RM | 97.0 | 93.7 | 97.9 | 100 | 90.0 | 73.0 | 83.0 | 32.1 | 21.7 | 46.8 | 26.1 | 78.0 | 99.0 | 95.5 | 81.6 | 50.0 | 47.7 | 82.3 | 83.5 | 86.6 | 79.9 | 81.1 | 83.5 | 71.7 |
| DR $\rho = 0.1$ | 99.0 | 91.6 | 97.9 | 100 | 95.0 | 75.7 | 83.0 | 34.3 | 20.7 | 42.6 | 21.7 | 79.0 | 99.0 | 95.5 | 82.0 | 50.7 | 55.5 | 78.7 | 83.5 | 83.5 | 78.7 | 84.1 | 79.3 | 72.7 |
| DR $\rho = 0.2$ | 99.0 | 91.6 | 97.9 | 100 | 92.5 | 75.7 | 81.0 | 31.3 | 21.7 | 40.4 | 26.1 | 78.0 | 99.0 | 96.1 | 79.2 | 51.5 | 61.7 | 83.5 | 83.5 | 85.4 | 81.7 | 86.0 | 83.5 | 73.3 |
| DR $\rho = 0.4$ | 99.0 | 95.8 | 97.9 | 96.4 | 90.0 | 70.3 | 82.0 | 30.6 | 23.9 | 44.7 | 28.3 | 74.0 | 99.0 | 96.8 | 80.0 | 52.9 | 71.1 | 84.8 | 81.1 | 83.5 | 79.9 | 81.1 | 82.3 | 74.6 |
| DR $\rho = 0.6$ | 99.0 | 93.7 | 98.9 | 100 | 92.5 | 73.0 | 80.0 | 26.9 | 18.5 | 40.4 | 28.3 | 71.0 | 99.0 | 95.5 | 82.4 | 47.1 | 55.3 | 81.1 | 81.7 | 83.5 | 79.9 | 86.6 | 83.5 | 71.8 |
| DR $\rho = 0.8$ | 100 | 94.7 | 98.9 | 100 | 92.5 | 73.0 | 85.0 | 30.6 | 18.5 | 46.8 | 26.1 | 61.0 | 97.0 | 94.2 | 81.2 | 44.1 | 66.2 | 85.4 | 78.0 | 81.7 | 82.3 | 84.8 | 82.9 | 73.3 |
| DR $\rho = 1.0$ | 99.0 | 94.7 | 98.9 | 92.9 | 90.0 | 73.0 | 79.0 | 25.4 | 16.3 | 46.8 | 26.1 | 69.0 | 99.0 | 94.8 | 81.6 | 51.5 | 70.9 | 81.7 | 78.7 | 82.9 | 80.5 | 85.4 | 81.7 | 73.4 |
| Base model: `mistralai/Mistral-7B-Instruct-v0.2` | | | | | | | | | | | | | | | | | | | | | | | | |
| BTL RM | 98.0 | 95.8 | 98.9 | 100 | 95.0 | 81.1 | 91.0 | 41.8 | 50.0 | 68.1 | 54.3 | 86.0 | 100 | 96.8 | 90.4 | 59.6 | 51.0 | 92.7 | 93.3 | 92.1 | 92.7 | 92.1 | 89.0 | 79.3 |
| DR $\varepsilon = 0.1$ | 99.0 | 93.7 | 100 | 100 | 95.0 | 81.1 | 91.0 | 41.0 | 46.7 | 63.8 | 47.8 | 87.0 | 98.0 | 97.4 | 91.6 | 61.0 | 39.8 | 93.3 | 95.1 | 93.9 | 92.7 | 87.8 | 90.2 | 77.6 |
| DR $\varepsilon = 0.2$ | 97.0 | 94.7 | 100 | 100 | 97.5 | 83.8 | 93.0 | 35.1 | 40.2 | 57.4 | 45.7 | 86.0 | 99.0 | 97.4 | 90.8 | 59.6 | 57.9 | 88.4 | 93.9 | 93.3 | 90.9 | 87.8 | 92.7 | 78.8 |
| DR $\varepsilon = 0.4$ | 95.0 | 77.9 | 100 | 100 | 82.5 | 54.1 | 79.0 | 32.8 | 28.3 | 53.2 | 37.0 | 85.0 | 100 | 98.7 | 81.6 | 55.9 | 52.3 | 61.0 | 54.9 | 62.8 | 62.2 | 62.2 | 54.3 | 69.1 |
| DR $\varepsilon = 0.6$ | 94.0 | 76.8 | 93.7 | 89.3 | 87.5 | 56.8 | 75.0 | 28.4 | 22.8 | 29.8 | 30.4 | 87.0 | 99.0 | 96.1 | 73.6 | 58.1 | 63.8 | 51.8 | 53.0 | 48.2 | 57.9 | 54.3 | 48.2 | 66.8 |
| DR $\varepsilon = 0.8$ | 90.0 | 72.6 | 91.6 | 89.3 | 75.0 | 51.4 | 67.0 | 35.8 | 22.8 | 44.7 | 39.1 | 73.0 | 90.0 | 90.3 | 76.4 | 55.9 | 43.4 | 54.3 | 56.1 | 55.5 | 50.0 | 57.3 | 57.9 | 63.2 |
| DR $\varepsilon = 1.0$ | 91.0 | 74.7 | 83.2 | 89.3 | 77.5 | 59.5 | 68.0 | 39.6 | 31.5 | 42.6 | 39.1 | 80.0 | 94.0 | 86.4 | 78.8 | 46.3 | 64.7 | 59.8 | 54.9 | 52.4 | 53.7 | 51.8 | 57.9 | 66.4 |

## B.3. DPO

For training DPO we use the same hyperparameters as shown in Table 14, except using the KL coefficient $\beta = 0.1$.

*Table 11.* Evaluation of the reward models on RewardBench (Lambert et al., 2024). Trained on a 400K item subset **filtering out the hh-rlhf subset** of the Unified-Feedback dataset (Jiang et al., 2023) for 2 epochs using **TV dist.**

| Reward Model | Chat | | | | | Chat-Hard | | | | | | Safety | | | | | Reasoning | | | | | | | Avg. |
|---|---|---|---|---|---|---|---|---|---|---|---|---|---|---|---|---|---|---|---|---|---|---|---|---|
| | 1 | 2 | 3 | 4 | 5 | 6 | 7 | 8 | 9 | 10 | 11 | 12 | 13 | 14 | 15 | 16 | 17 | 18 | 19 | 20 | 21 | 22 | 23 | |
| BTL RM | 98.0 | 90.5 | 98.9 | 96.4 | 90.0 | 73.0 | 82.0 | 31.3 | 20.7 | 57.4 | 28.3 | 77.0 | 99.0 | 95.5 | 86.0 | 51.5 | 56.6 | 82.9 | 88.4 | 87.2 | 81.1 | 87.8 | 82.9 | 73.6 |
| DR $\rho = 0.02$ | 97.0 | 92.6 | 96.8 | 100 | 90.0 | 70.3 | 86.0 | 33.6 | 21.7 | 55.3 | 34.8 | 76.0 | 99.0 | 94.8 | 86.0 | 47.8 | 53.5 | 81.7 | 86.0 | 86.0 | 86.0 | 85.4 | 81.1 | 73.3 |
| DR $\rho = 0.1$ | 98.0 | 94.7 | 98.9 | 96.4 | 92.5 | 73.0 | 84.0 | 30.6 | 23.9 | 48.9 | 34.8 | 75.0 | 98.0 | 94.8 | 86.4 | 48.5 | 55.7 | 84.1 | 89.6 | 86.6 | 83.5 | 88.4 | 81.1 | 73.8 |
| DR $\rho = 0.2$ | 98.0 | 93.7 | 98.9 | 100 | 90.0 | 73.0 | 83.0 | 32.8 | 21.7 | 53.2 | 30.4 | 75.0 | 99.0 | 93.5 | 87.2 | 50.0 | 55.9 | 83.5 | 86.0 | 89.6 | 80.5 | 87.2 | 79.3 | 73.7 |
| DR $\rho = 0.4$ | 98.0 | 94.7 | 97.9 | 92.9 | 90.0 | 70.3 | 83.0 | 30.6 | 20.7 | 42.6 | 30.4 | 76.0 | 98.0 | 92.2 | 84.8 | 45.6 | 50.3 | 82.3 | 87.8 | 85.4 | 82.9 | 86.0 | 79.3 | 71.8 |
| DR $\rho = 0.6$ | 99.0 | 93.7 | 97.9 | 96.4 | 90.0 | 70.3 | 80.0 | 29.1 | 15.2 | 36.2 | 32.6 | 70.0 | 99.0 | 94.8 | 84.8 | 45.6 | 71.8 | 84.8 | 85.4 | 89.6 | 84.1 | 84.8 | 79.9 | 74.0 |
| DR $\rho = 0.8$ | 99.0 | 91.6 | 98.9 | 100 | 92.5 | 73.0 | 81.0 | 31.3 | 19.6 | 44.7 | 28.3 | 78.0 | 98.0 | 94.8 | 85.2 | 44.9 | 50.6 | 82.9 | 89.6 | 89.0 | 86.0 | 87.2 | 79.3 | 72.3 |
| DR $\rho = 1.0$ | 98.0 | 88.4 | 95.8 | 96.4 | 90.0 | 56.8 | 72.0 | 31.3 | 16.3 | 42.6 | 15.2 | 71.0 | 98.0 | 94.2 | 83.6 | 50.7 | 63.5 | 43.3 | 44.5 | 43.3 | 42.7 | 49.4 | 44.5 | 66.7 |

*Table 12.* Evaluation of the reward models on RewardBench (Lambert et al., 2024). Trained on a 400K item subset **filtering out the hh-rlhf subset** of the Unified-Feedback dataset (Jiang et al., 2023) for 2 epochs using $\chi^2$ **dist.**

| Reward Model | Chat | | | | | Chat-Hard | | | | | | Safety | | | | | Reasoning | | | | | | | Avg. |
|---|---|---|---|---|---|---|---|---|---|---|---|---|---|---|---|---|---|---|---|---|---|---|---|---|
| | 1 | 2 | 3 | 4 | 5 | 6 | 7 | 8 | 9 | 10 | 11 | 12 | 13 | 14 | 15 | 16 | 17 | 18 | 19 | 20 | 21 | 22 | 23 | |
| BTL RM | 96.0 | 92.6 | 96.8 | 100 | 87.5 | 70.3 | 81.0 | 33.6 | 19.6 | 48.9 | 30.4 | 75.0 | 99.0 | 93.5 | 87.2 | 44.9 | 57.0 | 84.1 | 83.5 | 87.2 | 83.5 | 88.4 | 82.3 | 73.0 |
| DR $\rho = 0.05$ | 99.0 | 93.7 | 97.9 | 96.4 | 90.0 | 70.3 | 83.0 | 32.8 | 20.7 | 46.8 | 30.4 | 77.0 | 99.0 | 93.5 | 88.0 | 47.8 | 61.7 | 81.7 | 86.0 | 82.3 | 81.1 | 89.6 | 82.3 | 74.1 |
| DR $\rho = 0.1$ | 97.0 | 92.6 | 97.9 | 100 | 92.5 | 73.0 | 82.0 | 32.8 | 22.8 | 46.8 | 30.4 | 76.0 | 98.0 | 93.5 | 85.6 | 45.6 | 58.8 | 81.7 | 84.8 | 84.8 | 84.1 | 86.6 | 81.7 | 73.4 |
| DR $\rho = 0.2$ | 98.0 | 94.7 | 98.9 | 100 | 90.0 | 70.3 | 88.0 | 35.1 | 22.8 | 44.7 | 30.4 | 73.0 | 100 | 95.5 | 86.4 | 47.1 | 61.1 | 82.9 | 89.0 | 87.2 | 83.5 | 85.4 | 79.9 | 74.5 |
| DR $\rho = 0.3$ | 98.0 | 90.5 | 98.9 | 96.4 | 87.5 | 59.5 | 76.0 | 24.6 | 13.0 | 53.2 | 23.9 | 76.0 | 96.0 | 89.6 | 79.2 | 46.3 | 61.1 | 54.3 | 56.1 | 57.3 | 59.1 | 59.8 | 48.8 | 67.5 |
| DR $\rho = 0.4$ | 95.0 | 86.3 | 91.6 | 92.9 | 62.5 | 29.7 | 53.0 | 14.9 | 8.7 | 40.4 | 13.0 | 50.0 | 94.0 | 88.3 | 76.4 | 29.4 | 64.7 | 18.3 | 23.8 | 20.1 | 21.3 | 26.2 | 21.3 | 56.5 |
| DR $\rho = 0.5$ | 97.0 | 84.2 | 93.7 | 92.9 | 80.0 | 54.1 | 66.0 | 17.2 | 12.0 | 29.8 | 21.7 | 63.0 | 96.0 | 90.9 | 72.8 | 39.0 | 55.5 | 23.8 | 28.0 | 31.1 | 33.5 | 31.1 | 36.0 | 59.3 |

*Table 13.* Evaluation of the reward models on RewardBench (Lambert et al., 2024). Trained on a 400K item subset **filtering out the chatbot_arena_conversations subset** of the Unified-Feedback dataset (Jiang et al., 2023) for 2 epochs using **TV dist.**

| Reward Model | Chat | | | | | Chat-Hard | | | | | | Safety | | | | | Reasoning | | | | | | | Avg. |
|---|---|---|---|---|---|---|---|---|---|---|---|---|---|---|---|---|---|---|---|---|---|---|---|---|
| | 1 | 2 | 3 | 4 | 5 | 6 | 7 | 8 | 9 | 10 | 11 | 12 | 13 | 14 | 15 | 16 | 17 | 18 | 19 | 20 | 21 | 22 | 23 | |
| BTL RM | 99.0 | 94.7 | 97.9 | 96.4 | 92.5 | 67.6 | 84.0 | 31.3 | 18.5 | 42.6 | 23.9 | 76.0 | 99.0 | 95.5 | 80.0 | 53.7 | 55.0 | 84.8 | 84.8 | 82.3 | 87.8 | 83.5 | 81.7 | 72.5 |
| DR $\rho = 0.02$ | 98.0 | 93.7 | 97.9 | 100 | 92.5 | 64.9 | 80.0 | 31.3 | 20.7 | 42.6 | 21.7 | 75.0 | 99.0 | 94.8 | 82.0 | 50.0 | 54.8 | 84.8 | 83.5 | 84.8 | 85.4 | 87.2 | 82.9 | 72.3 |
| DR $\rho = 0.1$ | 98.0 | 93.7 | 97.9 | 100 | 90.0 | 70.3 | 81.0 | 32.8 | 21.7 | 42.6 | 26.1 | 76.0 | 99.0 | 95.5 | 82.4 | 51.5 | 57.7 | 82.3 | 81.7 | 84.8 | 84.8 | 86.0 | 81.1 | 73.0 |
| DR $\rho = 0.2$ | 100 | 95.8 | 97.9 | 100 | 92.5 | 67.6 | 81.0 | 29.9 | 16.3 | 46.8 | 26.1 | 78.0 | 100 | 96.8 | 82.4 | 51.5 | 53.9 | 81.7 | 82.9 | 84.1 | 79.3 | 82.9 | 78.0 | 72.4 |
| DR $\rho = 0.4$ | 99.0 | 87.4 | 98.9 | 92.9 | 85.0 | 62.2 | 78.0 | 23.9 | 17.4 | 40.4 | 26.1 | 70.0 | 99.0 | 96.1 | 79.2 | 52.2 | 68.2 | 59.1 | 56.7 | 58.5 | 65.9 | 61.6 | 61.0 | 69.2 |
| DR $\rho = 0.6$ | 89.0 | 75.8 | 92.6 | 85.7 | 67.5 | 37.8 | 50.0 | 15.7 | 13.0 | 27.7 | 8.7 | 57.0 | 98.0 | 90.9 | 66.0 | 36.0 | 36.5 | 20.7 | 20.7 | 23.2 | 25.6 | 23.2 | 23.8 | 51.8 |
| DR $\rho = 0.8$ | 89.0 | 76.8 | 86.3 | 82.1 | 77.5 | 48.6 | 56.0 | 36.6 | 13.0 | 25.5 | 32.6 | 79.0 | 99.0 | 87.0 | 70.0 | 44.9 | 67.6 | 37.8 | 45.1 | 41.5 | 47.0 | 42.7 | 39.0 | 61.9 |
| DR $\rho = 1.0$ | 86.0 | 73.7 | 82.1 | 78.6 | 67.5 | 43.2 | 53.0 | 29.1 | 21.7 | 44.7 | 21.7 | 70.0 | 91.0 | 90.3 | 80.0 | 55.1 | 65.1 | 38.4 | 45.7 | 48.8 | 52.4 | 46.3 | 48.2 | 61.9 |

*Table 14.* Hyperparameters used for DPO & PPO experiments.

| Hyperparameter | Value |
| --- | --- |
| Base model | google/gemma-2b-it |
| Number of epochs | 1 |
| Learning rate $\eta$ | 1e-5 |
| KL coefficient $\beta$ | 0.02 |
| Torch dtype | bfloat16 |
| Attention implementation | flash_attention_2 |
| Per device train batch size | 8 |
| Learning rate scheduler type | cosine |
| Warmup ratio | 0.03 |
| Weight decay | 0 |
| Optimizer | adamw_hf |
| Maximum sequence length | 1024 |
| LoRA $r$ | 32 |
| LoRA alpha | 64 |
| LoRA dropout | 0.05 |
| $\lambda$ (lambda, for GAE) | 0.95 |
| Discount factor $\gamma$ (gamma) | 1 |
| Clip range (policy) | 0.2 |
| Clip range value | 0.2 |
| Local rollout forward batch size | 4 |
| Missing EOS penalty | 1 |

*Table 15.* Evaluation of the PPO trained policy on Unified-Feedback (ID, Jiang et al., 2023), HHH Alignment (OOD, Askell et al., 2021), and MT Bench (OOD, Zheng et al., 2023) datasets. Trained on a 5K item subset of the Unified-Feedback dataset. *Version A* and *version B* indicate different versions of DR PPO described in the text.

| Model | Unified-Feedback | HHH Alignment | MT Bench |
|---|---|---|---|
| Base model: `google/gemma-2b-it` | | | |
| BTL RM, Std. PPO | 59.3 | 66.2 | 59.8 |
| DR PPO (version A) $\rho = 0.1$ | 59.4 | 68.1 | 64.7 |
| DR PPO (version A) $\rho = 0.2$ | 59.8 | 67.1 | 59.8 |
| DR PPO (version A) $\rho = 0.4$ | 60.3 | 68.1 | 60.4 |
| DR PPO (version A) $\rho = 0.6$ | 61.1 | 69.0 | 59.8 |
| DR PPO (version A) $\rho = 0.8$ | 61.4 | 68.5 | 61.0 |
| DR PPO (version A) $\rho = 1.0$ | 59.8 | 66.7 | 60.0 |
| DR PPO (version B) $\rho = 0.1$ | 59.9 | 66.2 | 58.8 |
| DR PPO (version B) $\rho = 0.2$ | 60.3 | 68.1 | 59.6 |
| DR PPO (version B) $\rho = 0.4$ | 60.2 | 66.7 | 59.5 |
| DR PPO (version B) $\rho = 0.6$ | 61.0 | 67.1 | 61.8 |
| DR PPO (version B) $\rho = 0.8$ | 57.6 | 67.6 | 51.9 |
| DR PPO (version B) $\rho = 1.0$ | 60.8 | 66.2 | 60.2 |
| Base model: `mistralai/Mistral-7B-Instruct-v0.2` | | | |
| BTL RM, Std. PPO | 36.8 | 42.1 | 31.3 |
| DR PPO (version A) $\rho = 0.1$ | 48.4 | 57.4 | 57.8 |
| DR PPO (version A) $\rho = 0.2$ | 49.6 | 52.3 | 57.0 |
| DR PPO (version A) $\rho = 0.4$ | 42.8 | 37.0 | 44.3 |
| DR PPO (version A) $\rho = 0.6$ | 37.3 | 34.3 | 28.1 |
| DR PPO (version A) $\rho = 0.8$ | 47.8 | 51.9 | 54.0 |
| DR PPO (version A) $\rho = 1.0$ | 37.5 | 44.0 | 32.9 |

*Table 16.* Evaluation of PPO on RewardBench (Lambert et al., 2024). Trained on a 5K item subset of the Unified-Feedback dataset (Jiang et al., 2023) using TV dist.

| Model | Chat | | | | | Chat-Hard | | | | | | Safety | | | | | Reasoning | | | | | | | Avg. |
|---|---|---|---|---|---|---|---|---|---|---|---|---|---|---|---|---|---|---|---|---|---|---|---|---|
| | 1 | 2 | 3 | 4 | 5 | 6 | 7 | 8 | 9 | 10 | 11 | 12 | 13 | 14 | 15 | 16 | 17 | 18 | 19 | 20 | 21 | 22 | 23 | |
| Std. PPO | 94.0 | 65.3 | 98.9 | 35.7 | 47.5 | 59.5 | 54.0 | 14.9 | 14.1 | 48.9 | 21.7 | 6.0 | 34.0 | 30.5 | 73.6 | 18.4 | 14.1 | 81.1 | 77.4 | 77.4 | 75.0 | 77.4 | 77.4 | 48.7 |
| $\rho = 0.02$ | 94.0 | 67.4 | 97.9 | 35.7 | 50.0 | 56.8 | 58.0 | 17.2 | 13.0 | 46.8 | 17.4 | 7.0 | 37.0 | 31.8 | 72.4 | 17.6 | 15.9 | 76.8 | 78.0 | 81.7 | 76.8 | 76.8 | 75.6 | 49.3 |
| $\rho = 0.1$ | 93.0 | 74.7 | 97.9 | 60.7 | 50.0 | 56.8 | 60.0 | 14.9 | 17.4 | 44.7 | 17.4 | 3.0 | 36.0 | 26.0 | 74.8 | 18.4 | 38.3 | 73.8 | 73.2 | 72.0 | 70.7 | 75.0 | 76.2 | 52.3 |
| $\rho = 0.2$ | 94.0 | 67.4 | 98.9 | 39.3 | 52.5 | 62.2 | 58.0 | 17.2 | 12.0 | 46.8 | 19.6 | 6.0 | 36.0 | 30.5 | 72.4 | 19.1 | 17.4 | 81.1 | 81.7 | 81.1 | 78.7 | 80.5 | 76.8 | 50.0 |
| $\rho = 0.4$ | 94.0 | 67.4 | 98.9 | 39.3 | 52.5 | 59.5 | 53.0 | 17.9 | 17.4 | 46.8 | 21.7 | 5.0 | 34.0 | 29.2 | 72.4 | 17.6 | 15.2 | 81.7 | 81.7 | 84.1 | 77.4 | 77.4 | 79.3 | 49.6 |
| $\rho = 0.6$ | 94.0 | 68.4 | 97.9 | 32.1 | 50.0 | 56.8 | 55.0 | 15.7 | 15.2 | 38.3 | 19.6 | 2.0 | 31.0 | 26.6 | 75.6 | 16.2 | 16.1 | 78.7 | 80.5 | 81.1 | 79.9 | 79.3 | 78.7 | 48.8 |
| $\rho = 0.8$ | 93.0 | 65.3 | 97.9 | 46.4 | 50.0 | 59.5 | 54.0 | 14.2 | 14.1 | 40.4 | 17.4 | 4.0 | 38.0 | 28.6 | 74.4 | 17.6 | 32.2 | 67.1 | 63.4 | 72.6 | 67.7 | 70.7 | 70.7 | 49.6 |
| $\rho = 1.0$ | 84.0 | 60.0 | 93.7 | 42.9 | 52.5 | 59.5 | 54.0 | 13.4 | 14.1 | 40.4 | 26.1 | 4.0 | 34.0 | 31.8 | 77.2 | 18.4 | 44.5 | 66.5 | 67.7 | 67.1 | 67.1 | 66.5 | 68.3 | 50.2 |

*Table 17.* Evaluation of the DPO trained policy on Unified-Feedback (ID, Jiang et al., 2023), HHH Alignment (OOD, Askell et al., 2021), and MT Bench (OOD, Zheng et al., 2023) datasets. Trained on a 400K item subset of the Unified-Feedback dataset using TV dist. Base model used for training: `mistralai/Mistral-7B-Instruct-v0.2`.

| Model | Unified-Feedback | HHH Alignment | MT Bench |
|---|---|---|---|
| Base model: `mistralai/Mistral-7B-Instruct-v0.2` | | | |
| Std. DPO | 69.5 | 75.5 | 67.7 |
| DR DPO $\rho = 0.1$ | 69.7 | 77.3 | 67.9 |
| DR DPO $\rho = 0.2$ | 70.1 | 78.7 | 69.4 |
| DR DPO $\rho = 0.4$ | 71.5 | 80.6 | 72.1 |
| DR DPO $\rho = 0.6$ | 72.2 | 80.1 | 71.9 |
| DR DPO $\rho = 0.8$ | 72.1 | 79.6 | 73.1 |
| DR DPO $\rho = 1.0$ | 69.6 | 70.8 | 73.2 |

*Table 18.* Evaluation of DPO on RewardBench (Lambert et al., 2024). Trained on a 20K item subset of the Unified-Feedback dataset (Jiang et al., 2023) using TV dist.

| Model | Chat | | | | | Chat-Hard | | | | | | Safety | | | | | Reasoning | | | | | | | Avg. |
|---|---|---|---|---|---|---|---|---|---|---|---|---|---|---|---|---|---|---|---|---|---|---|---|---|
| | 1 | 2 | 3 | 4 | 5 | 6 | 7 | 8 | 9 | 10 | 11 | 12 | 13 | 14 | 15 | 16 | 17 | 18 | 19 | 20 | 21 | 22 | 23 | |
| Std. DPO | 96.0 | 68.4 | 98.9 | 35.7 | 52.5 | 59.5 | 60.0 | 16.4 | 14.1 | 48.9 | 19.6 | 6.0 | 36.0 | 33.1 | 72.8 | 19.1 | 13.2 | 82.3 | 83.5 | 86.6 | 82.3 | 81.7 | 80.5 | 50.3 |
| $\rho = 0.02$ | 95.0 | 68.4 | 98.9 | 35.7 | 52.5 | 59.5 | 61.0 | 16.4 | 14.1 | 48.9 | 19.6 | 6.0 | 36.0 | 33.1 | 73.2 | 19.1 | 13.2 | 81.7 | 84.8 | 84.1 | 79.9 | 82.3 | 79.9 | 50.2 |
| $\rho = 0.1$ | 96.0 | 68.4 | 98.9 | 39.3 | 52.5 | 59.5 | 61.0 | 14.9 | 14.1 | 48.9 | 19.6 | 6.0 | 37.0 | 33.8 | 73.2 | 19.1 | 13.6 | 79.3 | 86.6 | 82.9 | 80.5 | 82.3 | 80.5 | 50.4 |
| $\rho = 0.2$ | 96.0 | 70.5 | 98.9 | 42.9 | 52.5 | 59.5 | 61.0 | 14.9 | 14.1 | 51.1 | 19.6 | 6.0 | 38.0 | 35.1 | 73.6 | 19.1 | 13.6 | 82.3 | 85.4 | 85.4 | 84.1 | 81.7 | 79.9 | 50.9 |
| $\rho = 0.4$ | 97.0 | 64.2 | 100.0 | 42.9 | 52.5 | 56.8 | 63.0 | 12.7 | 10.9 | 51.1 | 17.4 | 6.0 | 35.0 | 37.7 | 73.2 | 23.5 | 13.4 | 70.1 | 75.6 | 70.7 | 68.9 | 70.1 | 72.6 | 49.0 |
| $\rho = 0.6$ | 96.0 | 66.3 | 100.0 | 46.4 | 55.0 | 56.8 | 61.0 | 14.2 | 13.0 | 44.7 | 17.4 | 6.0 | 40.0 | 35.7 | 72.8 | 21.3 | 18.3 | 73.2 | 72.0 | 70.1 | 69.5 | 73.8 | 73.2 | 49.8 |
| $\rho = 0.8$ | 97.0 | 72.6 | 100.0 | 57.1 | 62.5 | 62.2 | 60.0 | 16.4 | 14.1 | 36.2 | 19.6 | 5.0 | 40.0 | 39.0 | 74.0 | 20.6 | 31.3 | 68.9 | 68.9 | 70.1 | 65.2 | 63.4 | 66.5 | 52.0 |
| $\rho = 1.0$ | 91.0 | 72.6 | 95.8 | 53.6 | 62.5 | 56.8 | 55.0 | 26.9 | 13.0 | 34.0 | 28.3 | 4.0 | 48.0 | 50.6 | 77.6 | 15.4 | 52.1 | 81.7 | 81.7 | 82.9 | 81.7 | 83.5 | 75.6 | 57.0 |

