# OpenReview forum: "Distributionally Robust Reinforcement Learning from Human Feedback"
_ICML.cc/2026/Conference — ICML 2026 regular_

### Official Review · Reviewer_8jxi · 2026-02-28

**Soundness:** 3
**Presentation:** 2
**Significance:** 2
**Originality:** 3
**Overall Recommendation:** 4
**Confidence:** 3

**Summary:**

This paper focuses on improving the robustness of reinforcement learning from human feedback under distribution shift. They consider the robustness in both reward modeling and policy optimization.

For RLHF, they show that under the linear reward, the robust reward estimation phase requires $O(1/\epsilon^2)$ iterations and $\tilde{O}(1/\epsilon^2)$ samples per iteration (Theorem 1). In the policy optimization, the proposed natural policy gradient method achieves a linear convergence rate $T=O(\log(1/\epsilon))$ while requiring $\tilde{O}(1/\epsilon^4)$ samples in each iteration (Theorem 2). The authors also proposed a robust version of direct preference optimization (DPO), with a rate of $O(1/\epsilon^2)$ iterations and $\tilde{O}(1/\epsilon^2)$ samples per iteration (Theorem 3).

Empirically, the methods are evaluated on Gemma-2B and Mistral-7B models trained on the Unified-Feedback dataset and evaluated on the out-of-distribution dataset, such as HHH-Alignment, and MT-Bench. Results show the improvement of the OOD performance, particularly for reasoning tasks (Tables 1–5; Sec. 5).

Overall, the paper gives a unified framework for improving OOD robustness in RLHF, with both theoretical analysis and experimental evaluation.

**Compliance With Llm Reviewing Policy:**

Affirmed.

**Final Justification:**

I think the paper has clear strengths in originality and technical interest, improving the robustness of reinforcement learning from human feedback under distribution shift. I am positive about this paper overall. The work offers a novel and technically interesting idea. The paper is also generally well written, and the empirical/theoretical results support the central claims reasonably well. I maintain the score 4.

**Key Questions For Authors:**

1. Could you elaborate on the novelty of the analysis, such as Theorem 2.

2. The paper focuses on the linear reward model. Can this approach extend to the general reward function case?

3. How sensitive is performance to the choice of the robustness radius $\rho$, and is this sensitivity good?

**Limitations:**

yes

**Strengths And Weaknesses:**

The paper is well-structured with separate formulations, algorithms, theoretical analysis, and experiments. The motivation regarding the distribution shift problem in RLHF is clearly stated. The work provides a well-defined formulation for both reward learning and policy optimization (Eq. (1)–(4), Sec. 3). The proposed algorithms are accompanied by convergence and sample complexity guarantees under stated assumptions (Theorems 1–3), addressing the OOD robustness problem in RLHF training. The theoretical analysis of robustness consideration also contributes new insights.

The empirical evaluation is conducted on 2B and 7B models and multiple benchmarks. The results show consistent OOD improvements and highlight substantial effects on reasoning subsets (Tables 1–5; Sec. 5).

---

> ### Author Rebuttal · Authors · 2026-03-31
>
> Dear reviewer, many thanks for your comments and insightful feedback. Below we answer your questions and address your concerns.
>
> ---
>
>  **Q1. Novelty of the analysis (Theorem 2**)
>
> Natural policy gradient (NPG) is a fundamental method in the theory of reinforcement learning. To the best of our knowledge, no prior work has neither proposed nor analyzed a distributionally robust version of NPG. Theorem 2 shows convergence of robust NPG with a definite bound on the number of iterations and batch size.
>
> In order to prove the theorem, we propose a non-trivial adaptation of proof of convergence of NPG. In particular, we adopt the proof of [a] who showed that NPG satisfies the recurrence $\Phi(\pi_{t+1}) \le (1-c) \Phi(\pi_t)$ where $\Phi(\cdot)$ is a potential function and $0<c<1$. The main novelty of our approach is to show that an approximate version of the recurrence relation holds when we use robust NPG, namely $\Phi(\pi_{t+1}) \le (1-c) \Phi(\pi_t) + O~(1/\sqrt{n})$. We utilize the dual characterization of the distributionally robust optimization to obtain the approximate recurrence relation. Afterwards, by choosing appropriate batch size $n$ and iteration $t$, we arrive at the result of theorem 2.
>
> [a] Convergence of entropy regularized natural policy gradient with linear function approximation. Cayci et. al., 2024
>
> ---
>
> **Q2. Generalization to nonlinear reward model**
>
> This is an interesting question and thanks for the suggestion! Our results for robust reward estimation naturally generalizes for nonlinear and nonconvex reward models. The main observation is that the bias of the minibatch gradient is independent of the choice of the reward model.
>
> For a general nonlinear reward model, Lemma 1 in the Appendix generalizes as long as the reward function is bounded. Algorithm 1 performs a biased gradient descent and this implies convergence to a stationary point of the distributionally robust objective. The linearity of the reward function is only used to guarantee convergence to global optima of algorithm 1. For a general nonconvex reward function, convergence to a stationary point is the best one can guarantee. However, with additional structural assumptions (e.g. standard Polyak-Lojasiewicz condition) we can obtain convergence to global optima.
>
> The proof of theorem 2 (robust policy optimization) doesn’t require linearity of reward functions, but uses the assumption that the policy is loglinear. In fact, we only require that the policy parametrization is smooth in the parameter. Therefore, we believe the proof can be generalized as long as the policy representation satisfies smoothness. We will add the discussion in the updated version of the paper.
>
> ---
>
> **Q3. Sensitivity with respect to the robustness radius ($\rho$)**
>
> Throughout our experiments, we have observed that the out-of-distribution (OOD) performance first improves as the robustness radius ($\rho$) increases, reaches a maxima for an intermediate value of $\rho$, and then decreases as $\rho$ approaches $1$. This is expected since for smaller values of $\rho$ no robustness is enforced and for larger values of $\rho$, robustness is enforced for almost all distributions leading to a decrease in accuracy. However, the performance is stable around an intermediate value of $\rho$ (e.g. $\rho \in \\{0.2,0.6\\}$ for robust DPO).
>
> We thank the reviewer again for reviewing our paper, as well as providing constructive feedback that helped us further clarify the effectiveness of our approach. We are happy to answer any additional comments or questions.
>
> ---

---

> > ### Author Rebuttal · Reviewer_8jxi · 2026-04-02
> >
> > Thanks for the detailed response. The response resolved my concerns and I maintain my original “Accept” score.

---

### Official Review · Reviewer_jsoL · 2026-03-09

**Soundness:** 3
**Presentation:** 3
**Significance:** 2
**Originality:** 2
**Overall Recommendation:** 4
**Confidence:** 3

**Summary:**

This paper studies out-of-distribution robustness in RLHF under prompt distribution shift. The authors formulate distributionally robust versions of reward-model learning, policy optimization, and DPO by optimizing against worst-case distributions inside a total-variation uncertainty set around the training distribution. They propose minibatch reweighting algorithms for robust reward estimation, robust policy optimization, and robust DPO, provide convergence and sample-complexity results under linear reward and log-linear policy assumptions, and evaluate the resulting methods on Unified-Feedback training data with RewardBench, HHH-Alignment, and MT-Bench as downstream evaluations. The proposed method improve both reward-model quality and OOD policy performance.

**Compliance With Llm Reviewing Policy:**

Affirmed.

**Final Justification:**

The rebuttal addressed my concerns and therefore I maintain my positive score.

**Key Questions For Authors:**

1. The theory centers on a robust natural policy gradient algorithm, but the experiments use PPO. How much of the empirical improvement survives when the implementation more closely matches the theory, and how should readers interpret the current guarantees for the PPO-based method?
2. How does the method compare experimentally against prior robust RLHF or robust DPO baselines, rather than only standard PPO/DPO?   This paper could benefit from including following baselines in comparison to strengthen the experiment.
   - **Xu et al. (2025)**, "Distributionally robust direct preference optimization" — which proposes DRO-DPO under Wasserstein and KL uncertainty sets;
   - **Wu et al. (2024a)**, "Towards robust alignment of language models: Distributionally robustifying direct preference optimization" — a DR-DPO variant focused on robustness to erroneous preference pairs
3. In reward models section, the author mentioned that "We observe an increase in performance for columns 1 (+23.4%), 2 (+3.1%), 5 (+2.4%), and 6 (+5.5%) in the reasoning task." (line 346-353), is there some explanation for performance gains concentrated on reasoning subsets (Table 3)? Meanwhile, for column 3, 4, 7, the baseline RM is performing quite strong. If the authors can provide a mechanistic explanation for the gains and loss, that would make the empirical results substantially more compelling.

**Limitations:**

The paper could benefit from including discussions for how distribution shift is measured, as mentioned in above.

**Strengths And Weaknesses:**

### Soundness
Strengths: The paper tackles a real and important failure mode of RLHF systems, namely degradation under prompt-distribution shift. The theoretical section provides explicit convergence/sample-complexity statements for robust reward learning, robust DPO, and a robust natural-policy-gradient variant. Empirically, the reward-model results are reasonably coherent, and the improvements on some reasoning subsets are large enough to suggest the method is doing something meaningful.

Weaknesses:
* There's certain soundness gap between the theory and the experiments. The main policy-learning theory is derived for a reweighted natural policy gradient algorithm under restrictive assumptions, but the empirical section uses a PPO-style implementation instead. The paper also lacks direct experimental comparisons to the most relevant robust RLHF or robust DPO baselines.
* The OOD experimental setup is relatively weak: the paper equates "different dataset" with "out-of-distribution" (train on Unified-Feedback, evaluate on HHH-Alignment / MT-Bench / RewardBench) without quantitatively characterizing the actual distributional gap in the TV-distance sense that the theory uses. There is no measurement of how far the evaluation distributions are from the training distribution, so it is unclear whether the robustness radius $\rho$ is well-calibrated to the real shift magnitude. A stronger setup would explicitly measure or control the shift magnitude and show that the optimal $\rho$ scales meaningfully with it.

### Presentation
Strengths: The paper is generally easy to follow. The three components are clearly separated, and the tables provide a usable view of where gains do and do not appear. The additional reasoning-subset breakdown is particularly useful because it identifies where the method seems most helpful.

### Significance
Strengths: OOD robustness in RLHF is important, and a unified DRO treatment across reward learning, policy optimization, and DPO is useful. If the empirical results generalize beyond the current setup, the paper could influence how practitioners think about robust alignment pipelines.

### Originality
Strengths: Applying minibatch DRO-style reweighting across multiple RLHF stages is a meaningful contribution, especially together with the robust DPO and policy-learning formulations.

---

> ### Author Rebuttal · Authors · 2026-03-31
>
> Dear reviewer, many thanks for your comments and insightful feedback. Below we provide answers to your questions and address your concerns.
>
> ---
>
> **Robust natural policy gradient (NPG) vs robust PPO**
>
> There are two motivations behind analyzing the performance of a distributionally robust variant of NPG. NPG is a fundamental method in the theory of reinforcement learning, and to the best of our knowledge, no prior work has neither proposed nor analyzed a distributionally robust version of NPG.
>
> Second, PPO can be thought of as a computationally inexpensive, first-order approximation of NPG. Therefore, we believe the analysis of DR NPG will be valuable in the analysis of DR PPO. The analysis of standard PPO is usually complex and not yet fully understood. Hence we hope to tackle the analysis of DR PPO in another paper.
>
> ---
>
> **Comparison against other robust algorithms**
>
> Thanks for the suggestion and we agree that comparison against a similar baseline would make our result more convincing. After the reviews were announced, we compared our method with DrDPO [a], a distributionally robust variant of DPO. In particular, we implemented DrDPO with KL-divergence (see eq. 12 for the specific loss function). It takes two parameters as input - $\beta’$ (regularization) and noise-rate (the fraction of labels to be flipped during training). The next table shows the OOD performance of DrDPO for the HHH-Alignment dataset (when trained on 20K subset of Unified Feedback).
>
> | $\beta’$ | noise-rate  | Accuracy  |
> |-----------|--------------|-------------|
> | 0.5	        | 0.0		    | 56.1	       |
> | 1.0	        | 0.0		    | 60.2	       |
> | 2.0	        | 0.0		    | 60.2	       |
> | 0.5	        | 0.1		    | 60.6	       |
> | 1.0	        | 0.1		    | 58.4	       |
> | 2.0	        | 0.1		    | 61.2	       |
> | 0.5	   | 0.2		| **63.3**	       |
> | 1.0	   | 0.2		| 58.1	       |
> | 2.0	   | 0.2		| 57.7	       |
>
>
> The best accuracy was 63.3%. This should be contrasted with the OOD performance of our method for a range of values of $\rho$. Our method achieves at least 5% higher accuracy.
>
> | $\rho$   | Accuracy   |
> |----------|--------------|
> | 0.2	   | 69.0	|
> | 0.4	   | 68.5	|
> | 0.6	   | 69.3	|
>
> We will be happy to provide a detailed comparison (including MTBench and larger models) in the full version of the paper.
>
> ---
>
> **Improved Performance on Reasoning Subsets**
>
> Thank you for the question and the suggestion! There is an intrinsic connection between reasoning and out-of-distribution robustness. A distributionally robust objective optimizes for worst-case performance over some uncertainty set, and forces the model to find features that work across perturbations of the data distribution. These tend to be causal features that correspond to actual reasoning rather than surface heuristics. We believe this is the primary reason the majority of the improvement is concentrated in the reasoning tasks.
>
> We think the mechanistic explanation of distributionally robust fine-tuning is a great suggestion. Preliminary analysis hasn’t revealed anything significant to report here, however, we believe a detailed mechanistic analysis of the DRO training is an exciting direction of future research, but is beyond the scope of the present paper.
>
> ---
>
> **The OOD experimental setup is relatively weak**
>
> We adopted the framework of [b] who used the similar framework to evaluate the OOD performance and generalization of reward model training. It would definitely be great to directly measure and control the distance between training data and target data. However, there are two main problems with text data. (1) Estimating distance is often difficult because of high-dimensional and sparse nature of the problem. (2) It is difficult to obtain a text dataset that is at a fixed distance from a training text corpora.
>
> ---
>
> We thank the reviewer again for reviewing our paper, as well as providing constructive feedback that helped us further clarify the effectiveness of our approach. We are happy to answer any additional comments or questions.
>
>
> **References**:
>
> [a] Towards Robust Alignment of Language Models: Distributionally Robustifying Direct Preference Optimization. Wu et. al. 2025.
>
> [b] Regularizing Hidden States Enables Learning Generalizable Reward Model for LLMs, Yang et. al. 2024.

---

> > ### Author Rebuttal · Reviewer_jsoL · 2026-04-03
> >
> > I thank the authors for the explanation. I therefore maintain my positive score.

---

### Official Review · Reviewer_pn7u · 2026-03-14

**Soundness:** 3
**Presentation:** 3
**Significance:** 3
**Originality:** 3
**Overall Recommendation:** 4
**Confidence:** 3

**Summary:**

This paper addresses the lack of robustness of current RLHF methods, in which model performance often degrades on downstream tasks whose data distributions differ from those used during training. To tackle this issue, the authors formalize distributionally robust variants of RLHF and DPO. They further propose practical algorithms based on minibatch gradient descent to implement these objectives. On the theoretical side, the paper provides convergence guarantees for the proposed methods. Experiments demonstrate that the approach can achieve better average performance on out-of-distribution (OOD) tasks than previous methods.

**Compliance With Llm Reviewing Policy:**

Affirmed.

**Key Questions For Authors:**

1. In the experiments, different values of $\rho$ lead to noticeably different performance across scenarios. In real deployment settings, how should $\rho$ be selected?

2. Is the proposed method essentially performing a form of implicit hard-example mining?

**Limitations:**

The paper would benefit from further discussion on whether the proposed methods remain effective and practical when applied to larger-scale models, as well as from a deeper analysis of the sensitivity to the choice of ( $\rho$ ) and how this issue might be mitigated in real-world applications.

**Strengths And Weaknesses:**

Strengths：
1. The paper proposes a distributionally robust formulation of RLHF implemented via a minibatch gradient-based algorithm. And further propose the methods of Robust Natural Policy Gradient and Robust Direct Preference Optimization.
2. The work provides a theoretical analysis of the proposed methods, including convergence guarantees as well as iteration and sample complexity.
3. Experimental results demonstrate that the proposed approach improves performance on out-of-distribution (OOD) tasks compared to standard RLHF/DPO baselines, indicating enhanced robustness under distribution shift.

Weaknesses：
Although the proposed method demonstrates improved performance in most experiments, I still have the following concerns:

1. While the paper provides a theoretical analysis of the iteration and sample complexity, these results rely on certain assumptions. In practical training scenarios, it remains unclear whether the proposed method introduces a high additional computational cost. This is a primary concern, and I would appreciate empirical comparisons (e.g., training time or resource overhead) to better assess its efficiency.

2. The experiments mainly compare against standard baselines, but do not include comparisons with more recent or closely related methods addressing robustness or generalization. Including such baselines would make the empirical validation more convincing.

3. The experimental results suggest that the method is relatively sensitive to the choice of the parameter \rho, with noticeably different outcomes under different values.

---

> ### Author Rebuttal · Authors · 2026-03-31
>
> Dear reviewer, thank you for your comments and insightful feedback. Below we provide answers to your questions and address your concerns.
>
> ---
>
> **Sensitivity of robustness radius $\rho$**
>
> Throughout our experiments, we observed that the out-of-distribution (OOD) performance first improves as the robustness radius ($\rho$) increases, reaches a maxima for an intermediate value of $\rho$, and then decreases as $\rho$ approaches $1$. This is expected since for smaller values of $\rho$ no robustness is enforced and for larger values of $\rho$, robustness is enforced for almost all distributions leading to a decrease in accuracy. However, the performance is stable around an intermediate value of $\rho$ (e.g. $\rho \in \\{0.2,0.6\\}$ for robust DPO).
>
> **Choice of robustness radius $\rho$**
>
> We suggest two methods to choose the parameter $\rho$ in practice. First, if a held-out set is available to evaluate the OOD performance then we can treat $\rho$ as a hyperparameter, and run a cross-validation over a grid (e.g. $\rho \in \\{0.01,0.1, 0.3, 0.7\\}$) evaluated on the held-out set to pick the best value of $\rho$. Otherwise, we can use the observation that various divergence functions (e.g. $\chi^2$) the DRO loss can be expressed as a variance regularized loss i.e. $\max_\theta E_{P}[r(\theta)] + O(\sqrt{\rho \cdot \text{Var}_P(r(\theta)})$. This problem is generally hard to solve. However, for each possible value of $\rho$ in a grid (e.g. $ \\{0.01,0.1, 0.3, 0.7\\}$) we can solve the DRO problem, and then pick the value of $\rho$ that optimizes mean plus square root of $\rho$ times variance. The choice of the optimal robustness radius is an active area of research [a] and we will include a discussion in the updated version of the paper.
>
> ---
>
> **Connections to hard-example mining**
>
> Yes, our approach is related to hard-example mining. In online hard-example mining, typically the loss is backpropagated through the top-k examples incurring the largest loss in a mini batch. However, we first figure out the worst-case distribution over the mini-batch and then the gradient is reweighted according to the worst-case distribution. Our approach is more general than the top-k hard example mining.
>
> ---
>
> **Comparison with alternate baseline**
>
> Thanks for the suggestion and we agree that comparison against similar baseline would make our result more convincing. After the reviews were announced, we compared our method with DrDPO [b], a distributionally robust variant of DPO. In particular, we implemented DrDPO with KL-divergence (see eq. 12 for the specific loss function). It takes two parameters as input - $\beta’$ (regularization) and noise-rate (the fraction of labels to be flipped during training). The next table show the OOD performance of DrDPO for the HHH-Alignment dataset (when trained on 20K subset of Unified Feedback).
>
> | $\beta’$ | noise-rate  | Accuracy  |
> |-----------|--------------|-------------|
> | 0.5	        | 0.0		    | 56.1	       |
> | 1.0	        | 0.0		    | 60.2	       |
> | 2.0	        | 0.0		    | 60.2	       |
> | 0.5	        | 0.1		    | 60.6	       |
> | 1.0	        | 0.1		    | 58.4	       |
> | 2.0	        | 0.1		    | 61.2	       |
> | 0.5	   | 0.2		| **63.3**	       |
> | 1.0	   | 0.2		| 58.1	       |
> | 2.0	   | 0.2		| 57.7	       |
>
>
> The best accuracy was 63.3%. This should be contrasted with the OOD performance of our method for a range of values of $\rho$. Our method achieves at least 5% higher accuracy.
>
> | $\rho$   | Accuracy   |
> |----------|--------------|
> | 0.2	   | 69.0	|
> | 0.4	   | 68.5	|
> | 0.6	   | 69.3	|
>
> We will be happy to provide a detailed comparison (including MTBench and larger models) in the full version of the paper.
>
> ---
>
> **Additional computational cost**
>
>  The only additional computation we need is the computation of worst-case distribution within a minibatch. This was done efficiently using solver like cvxpy. The actual clock time required to run robust RLHF/DPO/PPO methods are provided in appendix B, and they are very similar to non-robust versions.
>
> ---
>
> We thank the reviewer again for reviewing our paper, as well as providing constructive feedback that helped us further clarify the effectiveness of our approach. We are happy to answer any additional comments or questions.
>
> References:
>
> [a] Wasserstein Distributionally Robust Estimation in High Dimensions: Performance Analysis and Optimal Hyperparameter Tuning. Aolaritei et al. 2025.
>
> [b] Towards Robust Alignment of Language Models: Distributionally Robustifying Direct Preference Optimization. Wu et. al. 2025.

---

### Decision · Program_Chairs · 2026-04-30

**Decision:**

Accept (regular)

**Comment:**

This submission addresses the robustness issue of Reinforcement Learning from Human Feedback (RLHF) methods when facing distribution shifts in downstream tasks. The authors formulate distributionally robust optimization (DRO) variants of reward-based and reward-free RLHF training. Minibatch gradient descent algorithms are introduced, which enjoy theoretical convergence and sample complexity guarantees. Furthermore, the empirical evaluations successfully demonstrate improved average performance on out-of-distribution (OOD) datasets, with particularly notable gains in reasoning tasks.

The reviewers collectively recognized the significance of tackling RLHF robustness and appreciated the paper's unified DRO framework alongside its original theoretical analysis, specifically noting the convergence proofs for the robust natural policy gradient.

Initial weaknesses identified by the reviewers included a methodological gap between the theoretical natural policy gradient analysis and the practical PPO implementation, a lack of comparisons to recent robust baselines, and concerns regarding the model's sensitivity to the robustness radius parameter. The authors' rebuttal helped mitigate these concerns by contextualizing PPO as an approximation of the analyzed natural policy gradient, presenting convincing new empirical comparisons against DrDPO, and providing details about the stability of the robustness parameter at intermediate values.

Putting these aspects together, this paper stands out as a technically solid and valuable contribution to the field.